# Haploinsufficiency of the essential gene *Rps12* causes defects in erythropoiesis and hematopoietic stem cell maintenance

**Virginia Folgado-Marco**[1†], **Kristina Ames**[2,3†], **Jacky Chuen**[1], **Kira Gritsman**[2,3*], **Nicholas E Baker**[1*]

[1]Department of Genetics, Albert Einstein College of Medicine, Bronx, United States; [2]Department of Medical Oncology, Albert Einstein College of Medicine, Bronx, United States; [3]Department of Cell Biology, Albert Einstein College of Medicine, Bronx, United States

**\*For correspondence:**
kira.gritsman@einsteinmed.org (KG);
nicholas.baker@einsteinmed.edu (NEB)

†These authors contributed equally to this work

**Abstract** Ribosomal protein (Rp) gene haploinsufficiency can result in Diamond-Blackfan Anemia (DBA), characterized by defective erythropoiesis and skeletal defects. Some mouse Rp mutations recapitulate DBA phenotypes, although others lack erythropoietic or skeletal defects. We generated a conditional knockout mouse to partially delete *Rps12*. Homozygous *Rps12* deletion resulted in embryonic lethality. Mice inheriting the *Rps12*$^{KO/+}$ genotype had growth and morphological defects, pancytopenia, and impaired erythropoiesis. A striking reduction in hematopoietic stem cells (HSCs) and progenitors in the bone marrow (BM) was associated with decreased ability to repopulate the blood system after competitive and non-competitive BM transplantation. *Rps12*$^{KO/+}$ lost HSC quiescence, experienced ERK and MTOR activation, and increased global translation in HSC and progenitors. Post-natal heterozygous deletion of *Rps12* in hematopoietic cells using Tal1-Cre-ERT also resulted in pancytopenia with decreased HSC numbers. However, post-natal Cre-ERT induction led to reduced translation in HSCs and progenitors, suggesting that this is the most direct consequence of *Rps12* haploinsufficiency in hematopoietic cells. Thus, RpS12 has a strong requirement in HSC function, in addition to erythropoiesis.

## Editor's evaluation

This paper shows that haploinsufficiency of the ribosomal protein gene Rps12 in mice results in a number of phenotypes including defects in erythropoiesis, chronic pancytopenia, and loss of hematopoietic stem cell quiescence. This work will significantly add to the growing body of evidence that specific cell populations are particularly sensitive to global changes in mRNA translation. The manuscript contributes to our understanding of ribosome formation and function, mRNA translation, development, and stem cell biology.

## Introduction

In the cell, protein synthesis is one of the most energetically expensive processes, and both the specificity and overall level of translation are tightly regulated. The ribosome is the macromolecular machine tasked with translating mRNAs into proteins and, as such, plays an essential role in the physiology of the cell. Ribosomes are evolutionarily conserved ribonucleoprotein complexes composed of ribosomal RNA (rRNA) and Rp (*Yonath and Franceschi, 1998*; *Wilson and Doudna Cate, 2012*). They catalyze protein synthesis in all cell types, providing a supply line of steady-state levels of necessary cellular proteins (*Wilson and Doudna Cate, 2012*). The functional components of the ribosome are

highly conserved, and in higher eukaryotes consist of a small subunit (SSU: 40 S) and a large subunit (LSU: 60 S). These ribosomal subunits contain a total of 79 ribosomal proteins in eukaryotes, including 34 ribosomal proteins that are also conserved in prokaryotes (*Petibon et al., 2021*). In most cell types, ribosomal protein genes are among the most highly expressed genes (*Geiger et al., 2012*; *Ji et al., 2019*). Most ribosomal proteins are essential for ribosome biogenesis and function, which makes them essential for cell growth and proliferation (*de la Cruz et al., 2015*).

Given the importance of ribosomes, mutations in components of the ribosome or the ribosome biogenesis pathway in humans result in a group of diseases known as ribosomopathies. Despite the essential role of the ribosome in all cell types, this group of diseases is characterized by the presence of defects in specific tissues. Heterozygous loss of function mutations in many *Rp* genes leads to DBA, a congenital bone marrow failure syndrome characterized by macrocytic anemia, skeletal defects, and increased cancer risk. In DBA patients, mutations have been identified in 21 out of the 79 existing *Rp* genes, along with the GATA1 transcription factor (*Ulirsch et al., 2018*). Strikingly, in approximately 30–40% of DBA patient cases a mutation has not yet been identified. The fact that only a subset of all Rp genes has been found altered in DBA patients poses the question of whether mutations in any *Rp* gene can result in DBA and, if not, what would be the consequences for mutations in those *Rp* genes.

The generation and characterization of mice with mutations in *Rp* genes in recent years have begun to shed light on this question. Mutations in *Rp* genes of both the large and the small ribosomal subunits have been found to have similar phenotypes to those of DBA patients, such as impaired erythropoiesis, skeletal defects, and increased incidence of cancer, including some Rp not yet implicated in DBA (*Oliver et al., 2004*; *McGowan et al., 2008*; *Jaako et al., 2011*; *Terzian et al., 2011*; *Morgado-Palacin et al., 2015*; *Schneider et al., 2016*). However, erythropoietic defects are not always reported for mutants in *Rp* genes, including some that are implicated in human DBA (*Matsson et al., 2004*; *Watkins-Chow et al., 2013*; *Kazerounian et al., 2016*). Skeletal defects, but not impaired erythropoiesis, have been reported in mutants of *Rpl24* or *Rps7* (*Oliver et al., 2004*; *Watkins-Chow et al., 2013*).

In addition, a variety of other defects are reported in particular genotypes, ranging from embryonic lethality to brain defects, pigmentation defects, and defects in other aspects of hematopoiesis (*McGowan et al., 2008*; *Kondrashov et al., 2011*; *Terzian et al., 2011*; *Watkins-Chow et al., 2013*; *Morgado-Palacin et al., 2015*).

The *Rps12* gene, which is not yet implicated in DBA, reportedly has special functions in *Drosophila* that differ from those of most Rp. Heterozygous loss of 66 out of the 79 *Drosophila Rp* genes results in a 'Minute' phenotype, named for its small adult sensory bristles and also characterized by delayed development (*Marygold et al., 2007*). Additionally, 'Minute' *Rp*$^{+/-}$ are eliminated by wild-type (WT) neighboring cells when they are found together in developing tissues, by a process known as cell competition (*Morata and Ripoll, 1975*; *Clavería and Torres, 2016*; *Baker, 2020*). Remarkably, delayed development, reduced translation, cell competition, and other aspects of the 'Minute' phenotype depend on the Rps12 protein, which seems to be required for haploinsufficient effects of other *Rp* genes, suggesting that Rps12 acts as a sensor or reporter of deficits in other Rp. Accordingly, increasing the copy number of *Rps12* enhances these 'Minute' phenotypes caused by mutations in other *Rp* genes, whereas reducing the *Rps12* gene copy number suppresses them (*Kale et al., 2018*; *Boulan et al., 2019*; *Ji et al., 2019*). Rps12 forms part of the beak region of the 40 S small ribosomal subunit (SSU), near to the mRNA entry channel (*Rabl et al., 2011*). Interestingly, *Rps12* is one of the few *Rp* genes whose null mutation does not present a 'Minute' bristle phenotype in heterozygosis (*Marygold et al., 2007*; *Kale et al., 2018*). In mammals, it has been reported that *Rps12* deletions are frequent in diffuse large B cell lymphoma samples, and that Rps12 distribution in the ribosomes is altered under hypoxic conditions in the human embryonic kidney cell line, HEK293, resulting in changes of their translatome (*Derenzini et al., 2019*; *Brumwell et al., 2020*). Human Rps12 has also emerged as a candidate regulator of Wnt secretion in cancer cells (*Katanaev et al., 2020*). However, the phenotype of *Rps12* deletion in mammals has not been determined.

Protein synthesis regulation is important in stem cells. To maintain proper homeostasis, HSCs sustain the balance between a quiescent and an actively dividing state (*Cabezas-Wallscheid et al., 2017*). Quiescent HSCs require low rates of protein synthesis, and even HSCs exiting quiescence still exhibit significantly lower translation rates than more differentiated progenitors. Both increases and

decreases in protein synthesis levels can impair HSC function (*Signer et al., 2014*; *Hidalgo San Jose et al., 2020*).

The AKT/MTORC1 signaling pathway is one of the best-known signaling pathways that regulate translation, in part through the expression of ribosomal proteins and translation factors (*Fonseca et al., 2014*). Hyperactivation of AKT signaling is deleterious for normal HSC function, and results in increased HSC cycling, with depletion of the stem cell pool (*Yilmaz et al., 2006*; *Kharas et al., 2010*; *Lee et al., 2010*; *Magee et al., 2012*). Activation of AKT by stem cell factor (SCF) and other growth factors leads to the activation of MTOR, which results in the phosphorylation of the ribosomal protein S6 kinase 1 (S6K1) and the protein initiation factor 4E binding protein1 (4EBP1) (*Gentilella et al., 2015*). Phosphorylation of S6 by S6K1 at Serine 235/Ser236 is associated with increased protein translation (*Krieg et al., 1988*; *Roux and Topisirovic, 2018*). Additionally, phosphorylation of 4E-BP1 by MTOR at the Thr37 and Thr46 residues primes it for dissociation of 4E-BP1 from eIF4E, also activating translation (*Schalm et al., 2003*). Furthermore, another important pathway regulating growth and translation, the MEK/ERK pathway, has been shown to phosphorylate S6 at Serine 235/Ser236 promoting the translation preinitiation complex in mammalian cells (*Roux et al., 2007*).

To further explore the specific functions of *Rp* genes, their potential involvement in DBA, and the regulation of translation, we determined the phenotype of *Rps12* deletion in mice. We generated a conditional knock-out mouse, *Rps12*$^{flox/flox}$, which, when crossed to embryonically expressed Ella-Cre recombinase, allowed us to generate homozygous (*Rps12*$^{KO/KO}$) and heterozygous knock-out mice (*Rps12*$^{KO/+}$). We report that, while homozygous loss of *Rps12* is lethal early in embryogenesis, the viable heterozygous *Rps12*$^{KO/+}$ phenotype includes reduced body size, morphological defects, and, in some cases, hydrocephalus. Similar to DBA patients, and some other previously published *Rp* mouse mutants, *Rps12*$^{KO/+}$ mice present a block in erythroid maturation, lower red cell counts, and decreased spleen size. However, reduction in *Rps12* also leads to a striking reduction in the number of hematopoietic stem cells in the bone marrow, as well as significantly altered progenitor populations, leading to overall reduced bone marrow cellularity and a decreased ability of *Rps12*$^{KO/+}$ BM cells to repopulate the blood system, uncovering an impairment in HSC and progenitor function. Inducible, conditional heterozygous deletion of *Rps12* in adult hematopoietic cells also led to decreased HSC and progenitor cell numbers and pancytopenia. Interestingly, global translation was decreased after acute deletion, but increased in mice heterozygous for *Rps12* from fertilization. Our work has uncovered important roles for RpS12 in translation and signaling in hematopoietic cells.

## Results

### *Rps12* haploinsufficiency results in a pleiotropic phenotype, including delayed growth and increased mortality

To test the role of the Rps12 protein in a mammal, we used CRISPR gene editing to generate a mouse line with *LoxP* sites flanking exons 2 and 3 of the endogenous *Rps12* locus (*Rps12*$^{flox}$) (*Figure 1—figure supplement 1*). Excision of these two exons generates an allele that cannot produce functional RpS12 protein, since exon 2 contains the ATG translation initiation codon. We chose not to eliminate the entire *Rps12* locus, to avoid deleting the small nucleolar RNA genes *Snord100* and *Snora33*, which are located in introns 4 and 5, respectively (*Figure 1—figure supplement 1*). We crossed *Rps12*$^{flox/flox}$ mice to a line that expresses the Cre recombinase embryonically (EIIa-Cre) to obtain *Rps12* heterozygous knock-out (KO) mice (*Rps12*$^{KO/+}$) (*Figure 1A*). Unlike heterozygous null flies, which don't have any observable phenotype (*Marygold et al., 2007*; *Kale et al., 2018*), *Rps12*$^{KO/+}$ mice have reduced growth rates post-partum in comparison to their wild-type littermates (*Figure 1B and C*). Additional phenotypes include kinked tails, mild hyperpigmentation of the footpads, and an increased incidence of hydrocephalus (*Figure 1D, E and F*). Except for hydrocephalus, these phenotypes have also been found in some other Rp mutant mouse models (*Oliver et al., 2004*; *McGowan et al., 2008*; *Terzian et al., 2011*). Furthermore, *Rps12*$^{KO/+}$ mice have increased mortality, especially in early post-natal stages, most of which is associated with hydrocephalus or the inability to gain weight (*Figure 1G*).

To investigate if the KO allele of *Rps12* is lethal in the homozygous state, and whether *Rps12*$^{KO/+}$ animals have reduced growth during embryonic development, we crossed heterozygous *Rps12*$^{KO/+}$ male and female mice and analyzed the resulting embryos at stage E13.5 (we could not assess frequencies in pups at birth because *Rps12*$^{KO/+}$ females invariably died during labor). There were no

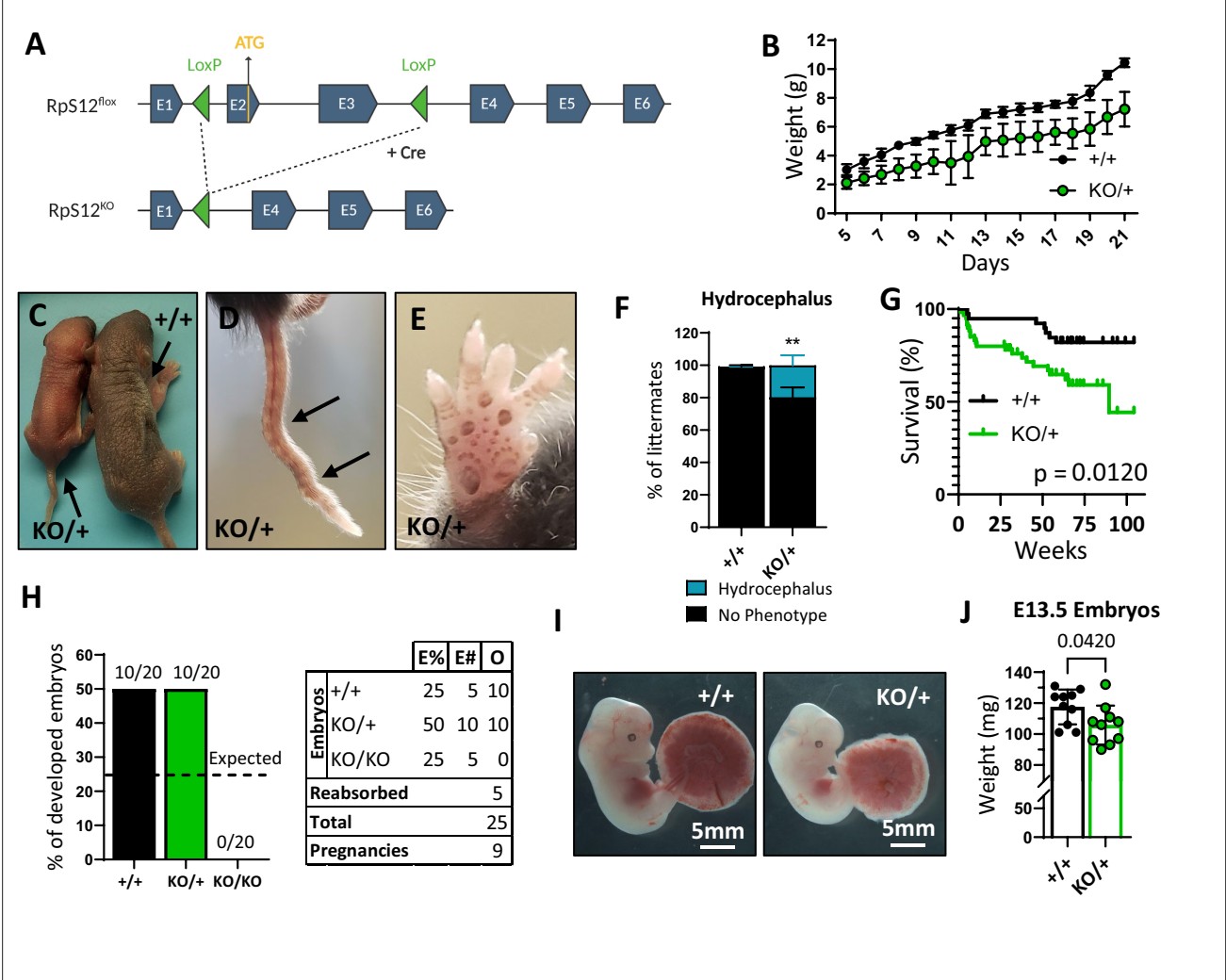

**Figure 1.** Loss of one copy of *Rps12* results in delayed growth, morphologic defects, and reduced viability. (**A**) Conditional *Rps12*[flox] transgenic knock-in has two loxP sites flanking exons 2 and 3, that are removed by Cre-ERT recombinase activity to generate *Rps12*[KO]. (**B**) Post-natal growth curve of *Rps12*[KO/+] and *Rps12*[+/+] littermates (+/+ n=8 and KO/+ n=11 pups). (**C**) Picture of 5-day-old *Rps12*[KO/+] and *Rps12*[+/+] littermates. (**D**) Representative picture of 'kinked' tail in *Rps12*[KO/+] mouse. (**E**) Representative picture of the anterior footpad hyperpigmentation in *Rps12*[KO/+]. (**F**) Quantification of the percentage of mice presenting hydrocephalus per litter (n=27 litters, two-way ANOVA p=0.0035). (**G**) Kaplan-Meier survival curves of *Rps12*[KO/+] and *Rps12*[+/+] littermates starting at day 5 of age (+/+ n=39 and KO/+ n=60, log-rank Mantel-Cox test p=0.012). (**H**) Embryo genotype segregation from crosses between *Rps12*[KO/+] male and female. Graph represents the percentage of developed embryos and the table shows the total numbers (E%=expected percentages, E#=expected numbers, O=observed numbers). (**I**) Representative pictures of E13.5 embryos with their placentas. (**J**) E13.5 embryo weights (n=10 on each genotype, unpaired t-test p=0.0420). Statistical analysis: quantifications represent mean +/− SEM, shown as the error bars.

The online version of this article includes the following source data and figure supplement(s) for figure 1:

**Source data 1.** Growth and embryo development.

**Figure supplement 1.** CRISPR gene editing and genotyping strategy for the generation of *Rps12*[flox] and *Rps12*[KO].

**Figure supplement 1—source data 1.** Blot for *Figure 1—figure supplement 1B*.

*Rps12*[KO/KO] specimens among the embryos obtained (*Figure 1H*), which led us to conclude that this genotype must be lethal prior to stage E13.5. Furthermore, *Rps12*[KO/+] embryos were smaller in size compared to their wildtype counterparts (*Figure 1I and J*). Therefore, these results indicate that *Rps12* is an essential gene, whose homozygous loss leads to early embryonic lethality, and heterozygous loss causes reduced growth starting in embryogenesis, in addition to other defects recognized post-partum.

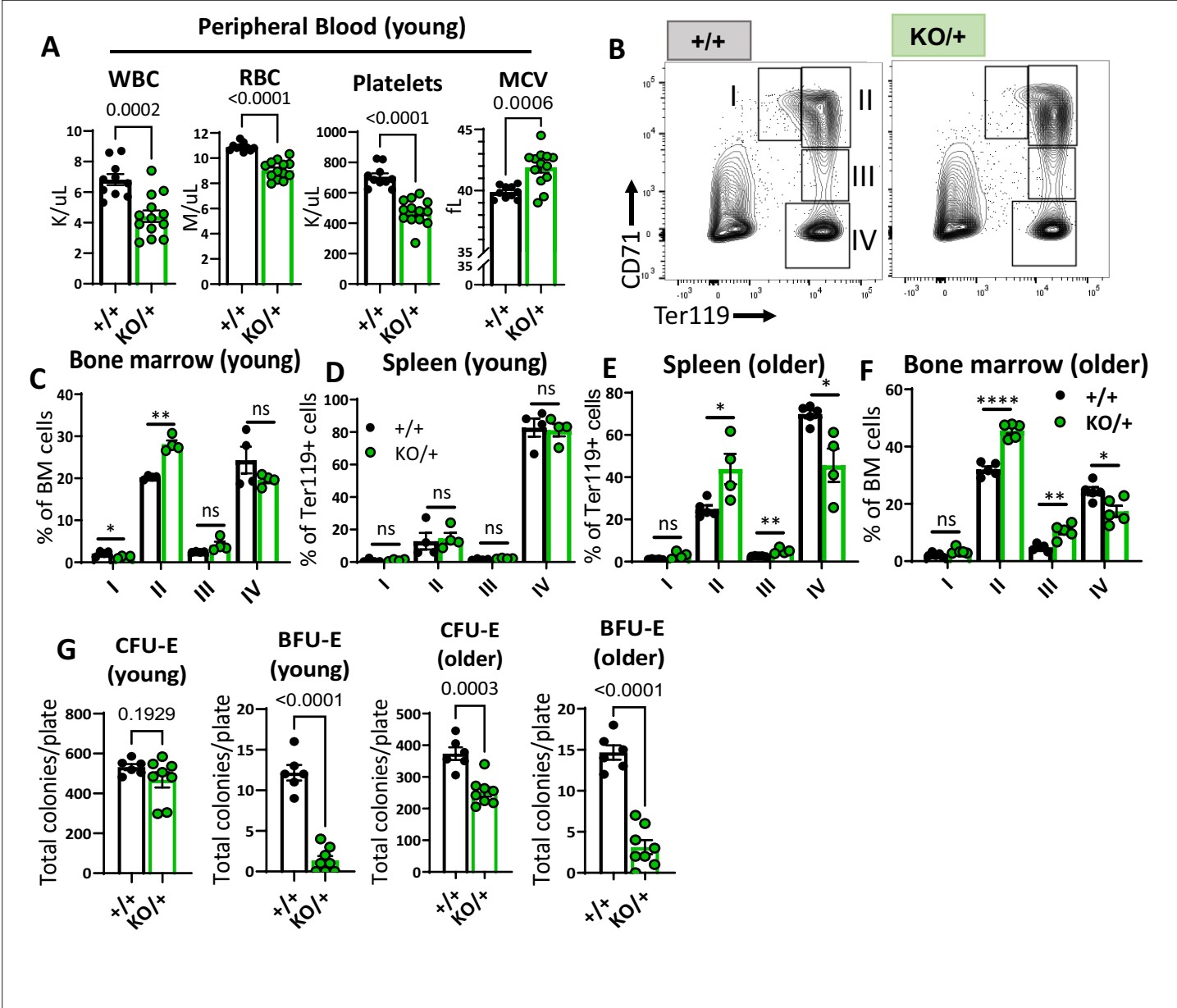

**Figure 2.** Haploinsufficiency of *Rps12* results in erythropoiesis defects that worsen with age. (**A**) Quantification of peripheral blood counts from young (6–8 weeks) littermates (+/+ n=10 and KO/+ n=13) (WBC=white blood cells, RBC=red blood cell, MCV=mean corpuscular volume). (**B**) Representative flow cytometry gating of bone marrow cells from 6- to 8-weeks-old mice of erythropoietic populations using Ter119 and CD71 markers. (**C, D, E, F**) Frequencies of erythroid progenitors in the bone marrow and spleen of young (6- to 7-weeks-old, +/+ n=4 and KO/+ n=4) and older (6- to 7-months-old, +/+ n=5 and KO/+ n=5) mice. (**G**) Total number of CFU-E and BFU-E colonies per plate (5 x 10⁵ bone marrow (BM) cells plated) in methylcellulose media supplemented with EPO (M3434) from young mice (6- to 7-weeks-old, +/+ n=4 and KO/+ n=4, each biological sample had two replicates) and older mice (6- to 7-months-old, +/+ n=4, and KO/+ n=4, each biological sample had two replicates). Statistical analysis: quantifications represent mean +/− SEM, shown as the error bars, when only two groups were being compared, unpaired t-test was performed, and for multiple comparisons one-way ANOVA analysis was used. *p<0.05, **p<0.01, ***p<0.001, ****p<0.0001.

The online version of this article includes the following source data for figure 2:

**Source data 1.** Erythropoiesis in younger and older mice.

## Heterozygous loss of *Rps12* results in erythropoiesis defects that worsen with age

We sought to understand if, similar to other Rp mutant mouse models, *Rps12* heterozygous mutants have anemia or defective erythropoiesis. Analysis of peripheral blood counts showed that 'young'

(6–8 weeks old) *Rps12*<sup>KO/+</sup> mice had a lower number of white blood cells (WBC), red blood cells (RBC), and platelets, a condition known as pancytopenia (*Figure 2A*). We also observed a high mean corpuscular volume (MCV), which is reminiscent of the macrocytic anemia seen in DBA patients.

To analyze erythropoiesis in *Rps12*<sup>KO/+</sup> mice, we used flow cytometry with the lineage markers Ter119 and CD71 on bone marrow and spleen cells (*Figure 2B*). These populations represent different maturation stages of the red blood cell production process, which we refer to as RI (CD71$^+$, Ter119$^-$, proerythroblasts), RII (CD71$^+$Ter119$^+$, basophilic erythroblasts), RIII (CD71$^{mid}$, Ter119$^+$, late basophilic and polychromatophilic erythroblasts), and RIV (CD71$^-$Ter119$^+$, orthochromatic erythroblasts) (*Socolovsky et al., 2001*). Bone marrow samples from young *Rps12*<sup>KO/+</sup> mice showed a defective transition between the RII and RIII stage cells, while the erythropoiesis in spleen populations was unchanged (*Figure 2C and D*). This impairment in erythropoiesis worsened with age, as samples from 'older' (6- to 7-month-old) *Rps12*<sup>KO/+</sup> mice had a higher accumulation of RII and RIII stage cells, while RIV population numbers were decreased in both spleen and bone marrow samples at this age (*Figure 2E and F*). To assess erythropoietic progenitor function, we performed colony-forming unit (CFU) assays in methylcellulose media optimized for the differentiation of erythroid progenitors. Consistent with the observed impairment of erythropoiesis, *Rps12*<sup>KO/+</sup> bone marrow cells generated fewer BFU-E colonies, indicating reduced erythroid progenitors (*Figure 2G*). Altogether, these results show that *Rps12* is required for erythroid differentiation, and demonstrate a role for *Rps12* in erythropoiesis, similar to what has been observed in mouse models of DBA genes like *Rpl11, Rps19, and Rps14* (*Jaako et al., 2011*; *Morgado-Palacin et al., 2015*; *Schneider et al., 2016*).

## *Rps12*<sup>KO/+</sup> mice have a striking reduction in hematopoietic progenitor populations, resulting in chronic pancytopenia

We were intrigued by the fact that *Rps12*<sup>KO/+</sup> mice have pancytopenia (*Figure 2A*), since this is not a common feature of DBA patients. Due to the general decrease of peripheral blood cell numbers in *Rps12*<sup>KO/+</sup> mice, we hypothesized that hematopoietic stem and progenitor cells (HSPCs) might be affected. Using flow cytometry analysis, we assessed the stem cell and progenitor populations in the bone marrow using previously defined markers (*Pietras et al., 2015*; *Figure 3A*). Indeed, compared to the controls, *Rps12*<sup>KO/+</sup> revealed a striking reduction in the numbers of long-term HSCs (LT-HSCs: Flk2$^-$CD48$^-$CD150$^+$ Lineage$^-$Sca1$^+$c-kit$^+$ (LSK)) and short-term HSCs (ST-HSCs: Flk2$^-$CD48$^-$CD150$^-$LSK) (*Figure 3B*). In addition, in *Rps12*<sup>KO/+</sup> bone marrow, the numbers of all hematopoietic progenitor populations were significantly reduced (*Figure 3C*; *Figure 3—figure supplement 1C*). Accordingly, compared to the WT littermates, young (6- to 8-weeks-old) *Rps12*<sup>KO/+</sup> mice had lower bone marrow cellularity, without obvious changes in the bone marrow architecture or dysplasia, and decreased spleen weights (*Figure 3D and E*, *Figure 3—figure supplement 1A, B*). Additionally, older (6- to 7-month-old) *Rps12*<sup>KO/+</sup> mice also had lower HSC numbers and frequency (*Figure 3F*, *Figure 3—figure supplement 1C and D*). Interestingly, we observed a partial recovery of some of the HSPC populations with age, such as multi-potent progenitors (MPP) 2 and 3, and the granulocyte-macrophage progenitors (GMP), as well as normalized overall BM cellularity, but not of spleen size (*Figure 3G, H and I*, *Figure 3—figure supplement 1D*). This, however, did not lead to improved blood counts (*Figure 3J*), indicating that HSPC function was not significantly improved with age.

To correlate these phenotypes with the expression levels of Rps12, rather than of the small RNAs *Snord100* and *Snora33*, we performed RT-PCR and Western-blot analysis of cKit-enriched bone marrow cells. As expected, RT-PCR results showed a highly significant decrease in the *Rps12* mRNA levels in *Rps12*<sup>KO/+</sup> cells (*Figure 3—figure supplement 2A*) but no difference in the levels of *Snord100* and *Snora33* (*Figure 3—figure supplement 2B*). Western-blot analysis also showed a strong trend toward decreased Rps12 protein levels in cKit + bone marrow cells (*Figure 3—figure supplement 2C*). We observed no changes to the ratio of the 28 S and 18 S rRNAs (*Figure 3—figure supplement 2D*), suggesting that steady-state SSU numbers were not depleted overall.

Lastly, since *Rps12* deletion resulted in decreased HSC and progenitor numbers, we assessed the self-renewal capacity of *Rps12*<sup>KO/+</sup> bone marrow cells. Plating assays in complete methylcellulose media showed a decreased clonogenic activity of *Rps12*<sup>KO/+</sup> bone marrow cells, as evidenced by the lower number of total colonies observed in the first round of plating (*Figure 3K*). Additionally, *Rps12*<sup>KO/+</sup> cells have reduced serial replating capacity, suggesting decreased self-renewal capacity (*Figure 3K*). Together, these results suggest that RpS12 plays an essential role in HSC function.

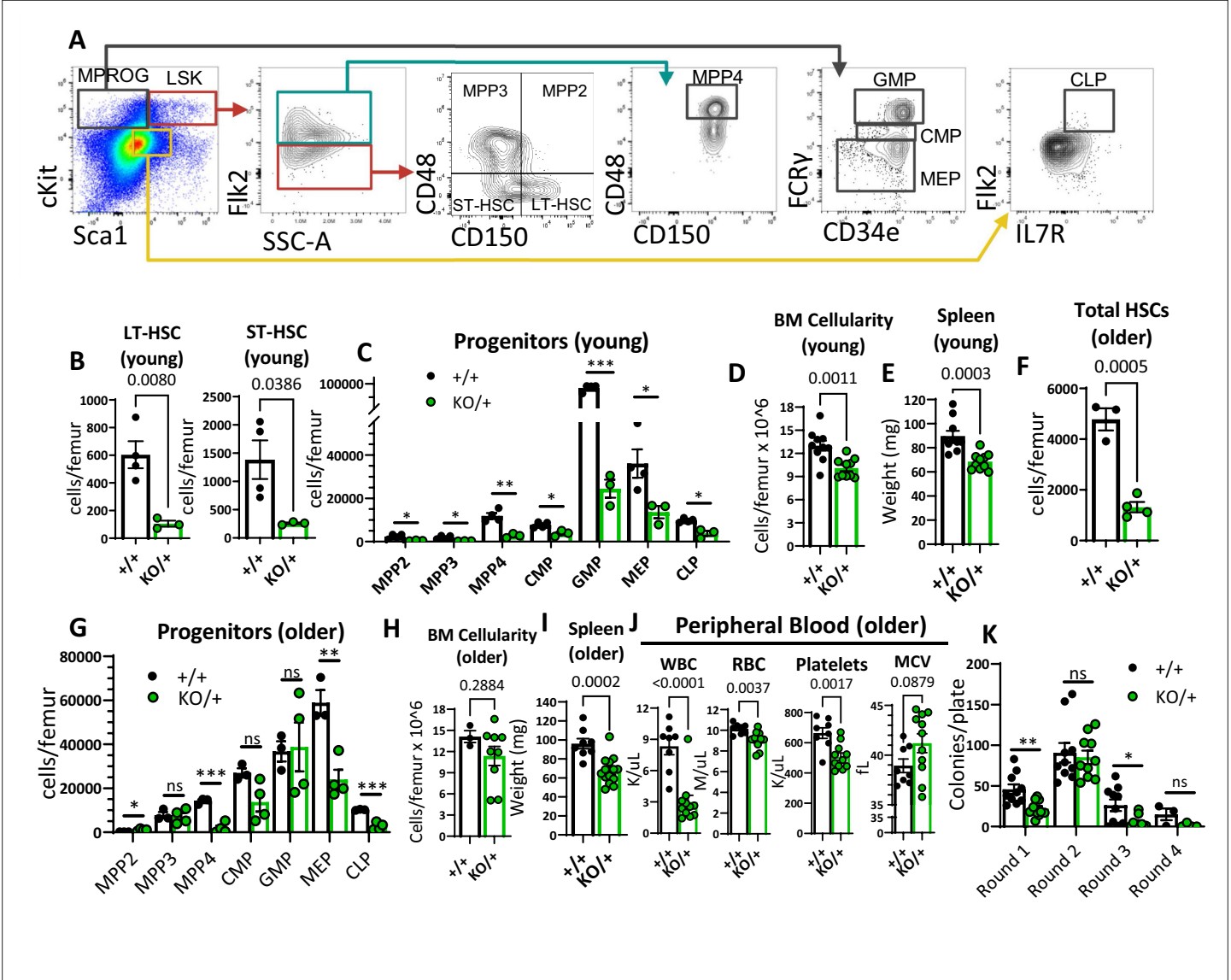

**Figure 3.** Reduced hematopoietic stem cells (HSCs) and other hematopoietic progenitor numbers in *Rps12*$^{KO/+}$ mice. (**A**) Representative gating strategy used to identify bone marrow populations of LSKs: long-term HSC (LT- HSC), short-term HSC (ST-HSC), multi-potent progenitors (MPP2, MPP3, MPP4), and myeloid progenitors (MPROG): common myeloid progenitor (CMP), granulocyte-monocyte progenitor (GMP), megakaryocyte-erythrocyte (MEP), and common lymphoid progenitor (CLP). (**B**) Total LT-HSCs and ST-HSCs per femur of young mice (6- to 8-weeks-old littermates, +/+ n=4, and KO/+ n=3). (**C**) Total number of cells per femur of indicated hematopoietic progenitor populations in young mice (6- to 8-weeks-old littermates, +/+ n=4, and KO/+ n=3). (**D**) Bone marrow cellularity is represented as cells per femur ×10$^6$ from young mice (6- to 7-weeks-old littermates, +/+ n=10, and KO/+ n=10). (**E**) Spleen weights of young (6- to 7-weeks-old, +/+ n=10, and KO/+ n=10) mice. (**F**) Total HSCs per femur of older mice (6- to 7-month-old, +/+ n=3, and KO/+ n=4). (**G**) Total number of cells per femur of indicated hematopoietic progenitor populations in older mice (6- to 7-month-old, +/+ n=3, and KO+ n=4). (**H**) Bone marrow cellularity is represented as cells per femur ×10$^6$ from older mice (older: 6- to 7-months-old, +/+ n=3, and KO/+ n=9). (**I**) Spleen weights of older (6- to 7-months-old, +/+ n=8, and KO/+ n=14) mice. (**J**) Quantification of peripheral blood counts from older mice (6- to 7 months old, +/+ n=8, and KO/+ n=11). (**K**) Total number of colonies per plate (1 × 10$^4$ bone marrow (BM) cells from 6- to 7-month-old mice plated in round 1 and 1 × 10$^4$ cells plated from the previous plate on each re-plating round) on each round of re-plating in complete methylcellulose media (+/+ n=5 and KO/+ n=5, 2 replicates per biological sample). Statistical analysis: quantifications represent mean +/− SEM, shown as the error bars, unpaired t-tests were performed to establish significance among populations between genotypes *p<0.05, **p<0.01, ***p<0.001, ****p<0.0001.

The online version of this article includes the following source data and figure supplement(s) for figure 3:

**Source data 1.** HSCs and progenitor population distributions.

**Figure supplement 1.** Heterozygous loss of *Rps12* in the hematopoietic cells.

**Figure supplement 1—source data 1.** Frequency of hematopoietic progenitors.

*Figure 3 continued on next page*

*Figure 3 continued*

**Figure supplement 2.** Decreased Rps12 and unaffected snoRNA expression levels in *Rps12*<sup>KO/+</sup> mice.

**Figure supplement 2—source data 1.** Blots of cKit + bone marrow cells.

**Figure supplement 2—source data 2.** RNA levels in ckit BM cells.

## Heterozygous loss of *Rps12* impairs the ability of HSCs to reconstitute peripheral blood

We assessed the self-renewal and differentiation properties of *Rps12*$^{KO/+}$ bone marrow cells (CD45.2+) in vivo by transplanting bone marrow into lethally irradiated B6.SJL mice (CD45.1+) (*Figure 4A*). Interestingly, compared to the *Rps12fl*$^{ox/flox}$ or *Rps12*$^{flox/+}$ controls, *Rps12*$^{KO/+}$ bone marrow recipients had decreased survival, with 5 out of 20 transplanted mice dying within the first 8 weeks in the *Rps12*$^{KO/+}$ group vs 0 out of 20 dying in the control group (*Figure 4B*). Whereas 100% of the control recipients were able to reconstitute the bone marrow in the long term (up to 20 weeks), only 55% of the *Rps12*$^{KO/+}$ recipients did so (*Figure 4C*). Furthermore, longitudinal analysis of donor chimerism in the peripheral blood revealed that compared to the controls, surviving *Rps12*$^{KO/+}$ transplant recipients had significantly decreased donor chimerism (%CD45.2+) in the B and T cell lineages, and a trend toward decreased chimerism in the myeloid lineage (*Figure 4D–G*). Together, this data suggests that bone marrow cells that lack RpS12 are deficient in hematopoietic repopulating capacity after lethal irradiation.

To assess *Rps12*$^{KO/+}$ bone marrow cell repopulation capacity under more stringent conditions, we performed competitive transplantation of control (*Rps12*$^{flox/+}$) or *Rps12*$^{KO/+}$ bone marrow (CD45.2$^+$) mixed with competitor WT bone marrow from B6.SJL mice (CD45.1$^+$) in a 1:1 ratio into lethally irradiated B6.SJL (CD45.1$^+$) recipient mice. Post-transplantation we monitored donor chimerism in the peripheral blood over time and analyzed the bone marrow chimerism at 20 weeks post-transplantation (*Figure 4H*). Compared to the controls, *Rps12*$^{KO/+}$ transplant recipients showed a striking decrease in the percentage of donor-derived bone marrow cells and of HSCs (*Figure 4I and J*), accompanied by a significant and persistent reduction in peripheral blood total donor chimerism in both myeloid and lymphoid lineages (*Figure 4K–N*). Together, these data suggest that *Rps12* haploinsufficiency leads to perturbed HSC self-renewal, resulting in ineffective hematopoiesis.

## The embryonic hematopoietic system is largely unaffected in *Rps12*$^{KO/+}$ animals

The striking reduction of hematopoietic progenitor numbers in *Rps12*$^{KO/+}$ adult bone marrow prompted us to investigate if this phenotype could be a consequence of defective HSC production in the fetal liver during embryogenesis. We analyzed fetal liver hematopoietic populations of E13.5 *Rps12*$^{+/+}$ and *Rps12*$^{KO/+}$ embryos because it has been shown that HSC numbers increase, and differentiation begins, between days 12 and 16 of embryogenesis in this organ (*Sugiyama et al., 2011*). First, we looked at the gross morphology and cellularity of the liver, neither of which were significantly different between the genotypes (*Figure 5A and B*). Next, we assessed different stages of erythropoiesis using Ter119 and CD71 markers as previously described in fetal liver (*Magee and Signer, 2021*). Most of the cells in population V were lost during staining, and, therefore, we did not include them in our analysis. Compared to the control embryos, *Rps12*$^{KO/+}$ embryos showed no apparent impairment in erythropoiesis in the fetal liver (*Figure 5C and D*). Finally, we analyzed the distribution of HSPCs in E13.5 embryos. Overall, we did not observe any significant changes in the frequencies of LT-HSCs (CD48$^-$CD150$^+$ LSK), ST-HSCs (CD48$^-$CD150$^-$ LSK), MPP (CD48$^+$ LSK), CMP, GMP, or MEP populations (*Figure 5E–H*). These results show that partial loss of *Rps12* does not affect embryonic hematopoiesis by E13.5. Therefore, the later HSPC deficiency is not a consequence of a defect in the embryonic specification.

## *Rps12*$^{KO/+}$ HSCs and some hematopoietic progenitors show higher translation, cycling, and apoptosis

There are several important factors that maintain the HSC pool, including quiescence, low translation levels, and cell survival. We, therefore, analyzed the distribution of HSPCs among the cell cycle stages defined by the DNA content (Hoechst) and the levels of Ki67 (*Figure 6A*). We observed a

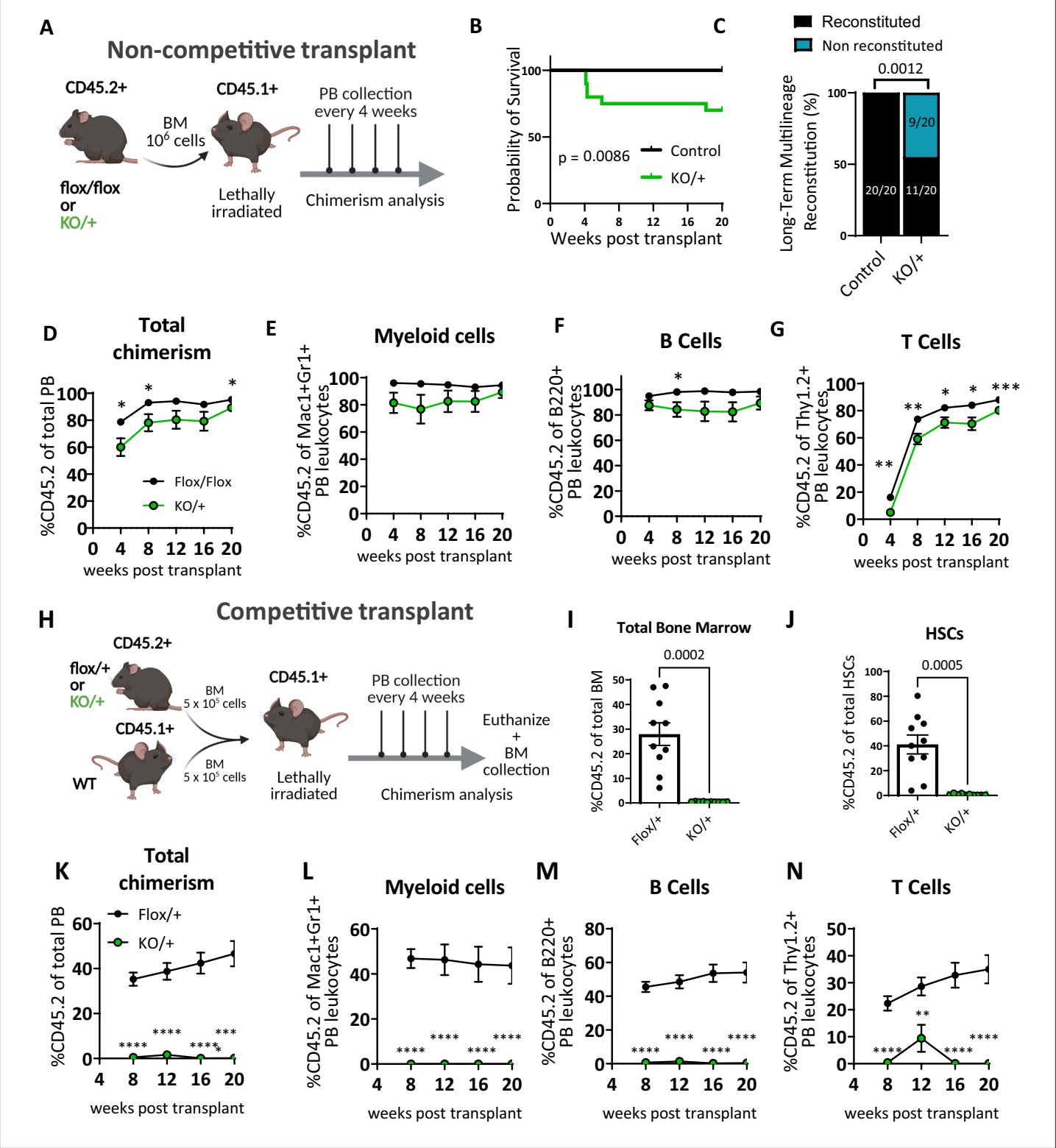

**Figure 4.** Heterozygous loss of *Rps12* impairs hematopoietic stem cells (HSCs) ability to reconstitute peripheral blood. (**A**) Non-competitive bone marrow (BM) transplant strategy testing the long-term reconstituting activity of *Rps12*^KO/+^ HSCs. $10^6$ bone marrow cells from *Rps12*^KO/+^ or *Rps12*^flox/flox^ samples (CD45.2+) were transplanted into lethally irradiated B6.SJL (CD45.1+) mice, peripheral blood chimerism was determined every 4 weeks. (**B**) Kaplan-Meier survival curves of mice transplanted with BM cells from *Rps12*^KO/+^ and control *Rps12*^flox/+^ or *Rps12*^flox/flox^ mice (control n=20 and KO/+ n=20 transplanted mice, the combination of two independent non-competitive transplants with 1 donor per genotype transplanted into 10 host mice

*Figure 4 continued*

each). (**C**) Frequency of recipient mice with long-term (20 weeks) multi-lineage reconstitution (≥0.5% in all three macrophages, B, and T cells) (control n=20 and KO/+ n=20 transplanted mice, the combination of two independent non-competitive transplants). (**D–G**) Peripheral blood donor-derived (**D**) total chimerism and (**E–G**) multi-lineage chimerism in non-competitively transplanted whole bone marrow (CD45.2+) recipients (*flox/flox* n=10 and *KO/+* n=10). (**H**) Schematic representation of the competitive bone marrow transplant. $5 \times 10^5$ cells from $Rps12^{KO/+}$ or $Rps12^{flox/+}$ donor bone marrow (CD45.2+) mixed with $5 \times 10^5$ competitor bone marrow cells from B6.SJL (CD45.1+) mice were injected into lethally irradiated B6.SJL (CD45.1+) mice. Chimerism in peripheral blood was determined every 4 weeks and bone marrow chimerism was analyzed at 20 weeks after transplant. (**I**) Total bone marrow chimerism and (**J**) HSCs donor-derived (CD45.2+) chimerism in the recipient bone marrow (*flox/+* n=10 and *KO/+* n=10 competitive-transplanted mice). (**K–N**) Donor-derived peripheral blood chimerism of competitively transplanted $Rps12^{KO/+}$ or $Rps12^{flox/+}$ bone marrow cells as described in (**H**). Non-competitive transplants were performed twice, using different controls: $Rps12^{flox/+}$ or $Rps12^{flox/flox}$. The competitive transplant was performed once, using $Rps12^{flox/+}$ mice as a control group. Statistical analysis: data represent mean +/− SEM, shown as the error bars, unpaired t-tests were performed to assess significance among populations between genotypes *p<0.05, **p<0.01, ***p<0.001, ****p<0.0001.

The online version of this article includes the following source data for figure 4:

**Source data 1.** Bone marrow transplants.

lower proportion of $Rps12^{KO/+}$ HSCs in the G0 stage of the cell cycle, and a significantly increased proportion in the active cycling phases G1 and S/G2/M (**Figure 6B**). Similar results were observed in MPP2/3 (LSK, Flk2⁻, CD48⁺), MPP4, and myeloid (MPROG) and common lymphoid progenitors (CLP) (**Figure 6B**). These results show that compared to the control, $Rps12^{KO/+}$ HSCs are significantly less quiescent, with a higher proportion of HSCs and progenitors actively cycling.

Cell cycle activation generally requires translation, but previous studies have reported a generalized decrease in global translation in some Rp mutants, including in HSPC, despite a decrease in HSC quiescence (**Oliver et al., 2004**; **Signer et al., 2014**; **Schneider et al., 2016**). To assess the global translation levels of each HSPC population in *Rps12* heterozygous mice using flow cytometry, we performed an ex vivo assay on freshly isolated HSCs and progenitors using the puromycin analog o-propargyl puromycin (OPP) as previously described (**Signer et al., 2014**). Unexpectedly, compared to the $Rps12^{+/+}$ controls, $Rps12^{KO/+}$ HSCs and multipotent progenitor cell populations all showed increased levels of global translation (**Figure 6C and D**). The difference was especially remarkable in HSCs. Interestingly, compared to the controls, $Rps12^{KO/+}$ myeloid progenitors did not exhibit differences in OPP intensity, and among different myeloid progenitor populations, only the megakaryocyte-erythrocyte progenitors (MEP) had a significant increase in OPP incorporation (**Figure 6E**). Thus, these data suggest that a decrease in RpS12 leads to an abnormal increase in global protein translation in immature bone marrow populations, including HSCs.

Cell death can deplete the HSC pool and can result from chronic HSC activation. We asked whether this reduction of HSCs in $Rps12^{KO/+}$ animals is due to an increase in apoptosis. Interestingly, compared to controls, $Rps12^{KO/+}$ animals have an increased number of apoptotic cells in bone marrow cytospins (**Figure 6F**). To quantify the level of apoptosis in the immunophenotypic populations in the bone marrow, we used the flow cytometry markers PI and Annexin V together with population-specific cell surface markers (**Figure 6G**). Our flow cytometry analysis confirmed a significant increase in apoptosis in Lineage⁻Sca1⁺c-Kit⁺ (LSK) cells, a population that contains HSCs and MPPs, but not in more mature myeloid progenitors (**Figure 6H and I**).

## $Rps12^{KO/+}$ HSPCs have overactivated MEK/ERK and AKT/MTOR signaling pathways

Because $Rps12^{KO/+}$ mutants have increased translation, we assessed the activity of the AKT/MTOR pathway, since it is known to regulate translation. Since the AKT/MTOR pathway is activated by stem cell factor (SCF), we determined the level of the AKT/MTOR pathway activation in the presence and absence of SCF, by assessing the phosphorylation levels of phospho-AKT (Ser 473) and the MTOR downstream effectors phospho-S6 (Ser235/236) and phospho-4E-BP1 (Thr37/46). Our results show that, in the more immature LSK population, which includes HSCs and MPPs, the levels of p-AKT, p-S6, and p-4E-BP1 were significantly elevated in $Rps12^{KO/+}$ animals compared to wild-type littermates, not only upon SCF stimulation, but even at the non-stimulated baseline (**Figure 7A–C**). Interestingly, this was not the case for the more mature myeloid progenitor cells, where the levels of p-AKT, p-S6, and p-4E-BP1 are comparable to the controls in both non-stimulated and SCF-stimulated conditions, and these cells also exhibited more normal translation rates (**Figure 7D–F**). Since phosphorylation

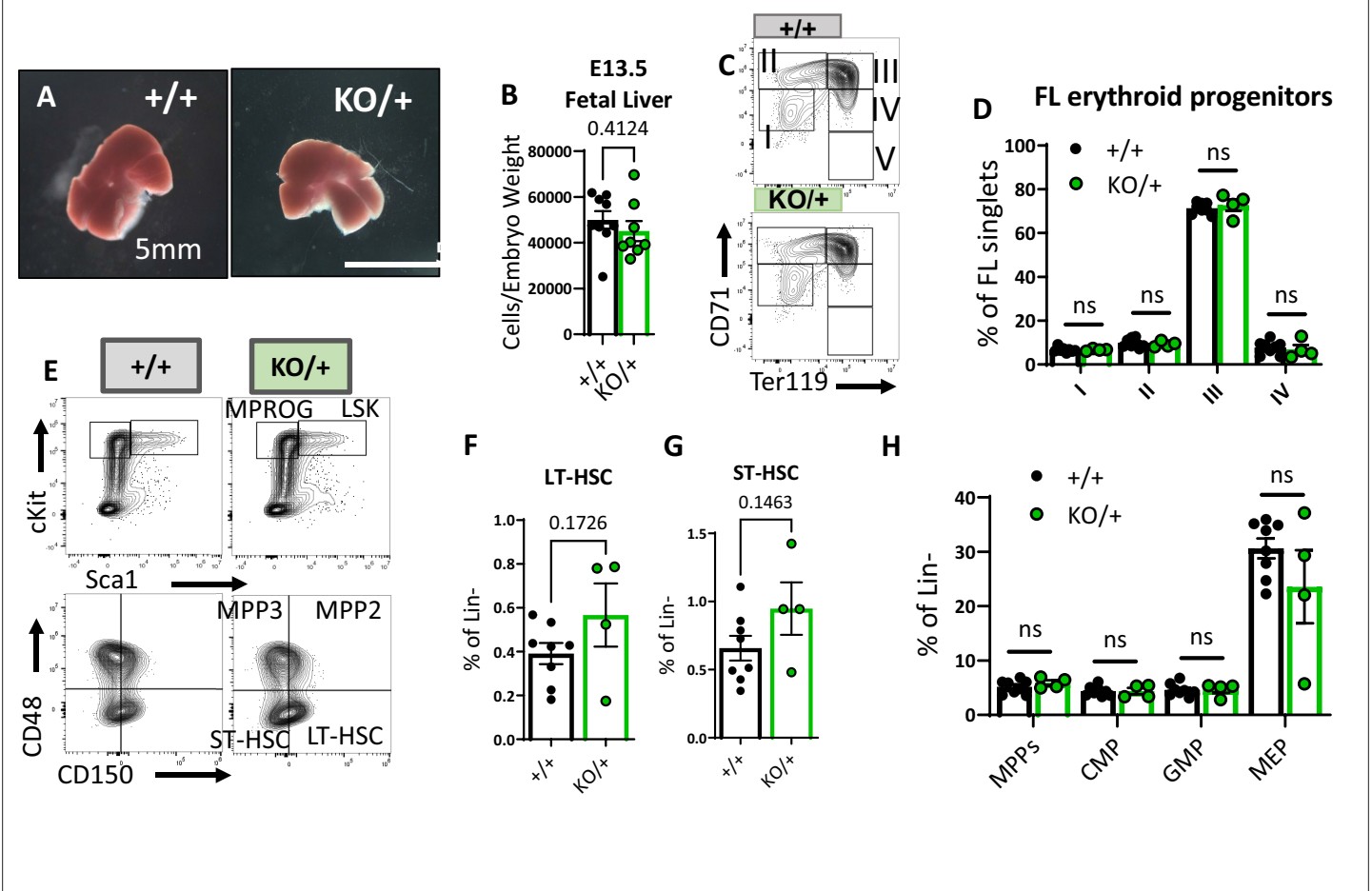

**Figure 5.** Embryonic hematopoietic system is largely unaffected in *Rps12*^KO/+^ animals. (**A**) Representative images of *Rps12*^+/-^ and littermate E13.5 fetal livers. (**B**) Quantification of the total number of cells per liver, normalized to embryo weight (+/+ n=9 and KO/+ n = 8). (**C**) Representative flow cytometry gating of erythropoietic populations using Ter119 and CD71 markers of fetal liver samples from E13.5 embryos. (**E**) Representative flow cytometry gating of Lin- (top) and LSK (bottom) populations in E13.5 fetal livers. (**F, G, H**) LT-HSCs, ST-HSCs, and indicated progenitor populations represented as percentages of the Lin- population in E13.5 fetal livers. (**D, F, G, H**) Biological samples are +/+ n=8 and KO/+ n=4. Statistical analysis: quantifications represent mean +/− SEM, shown as the error bars, unpaired t-tests were performed to establish significance among populations between genotypes *p<0.05, **p<0.01, ***p<0.001, ****p<0.0001.

The online version of this article includes the following source data for figure 5:

**Source data 1.** Fetal liver hematopoietic populations.

of S6 and 4EBP1 leads to increased translation, this data corroborates the increase in translation observed in the *Rps12*^KO/+^ LSK population (**Figure 6D**). Interestingly, more mature MPROG do not have increased translation (**Figure 6E**) and do not have increased activation of the AKT/MTOR pathway (**Figure 7D–F**). Together, these data suggest that activation of translation and the increase in AKT/MTOR signaling in *Rps12*^KO/+^ mutant cells are specific to HSCs and MPPs.

Additionally, compared to wild-type controls, *Rps12*^KO/+^ mutant animals also have increased phospho-ERK1 (Thr202/Tyr204) in the LSK and MPROG populations under SCF-stimulated and non-stimulated conditions (**Figure 7G and H**). Another regulator of translation is the eukaryotic initiation factor 2α (eIF2α), which is required for CAP-dependent translation initiation. Cells respond to several stress conditions by phosphorylating eIF2α, reducing global translation, and upregulating stress-response genes (**Wek et al., 2006**; **Sigurdsson and Miharada, 2018**). In skeletal muscle stem cells, eIF2α phosphorylation promotes quiescence and stem cell maintenance (**Zismanov et al., 2016**) Interestingly, compared to the control cells, *Rps12*^KO/+^ cKit+ bone marrow progenitor cells show decreased levels of p-eIF2α (**Figure 7I**), which also correlates with increased translation.

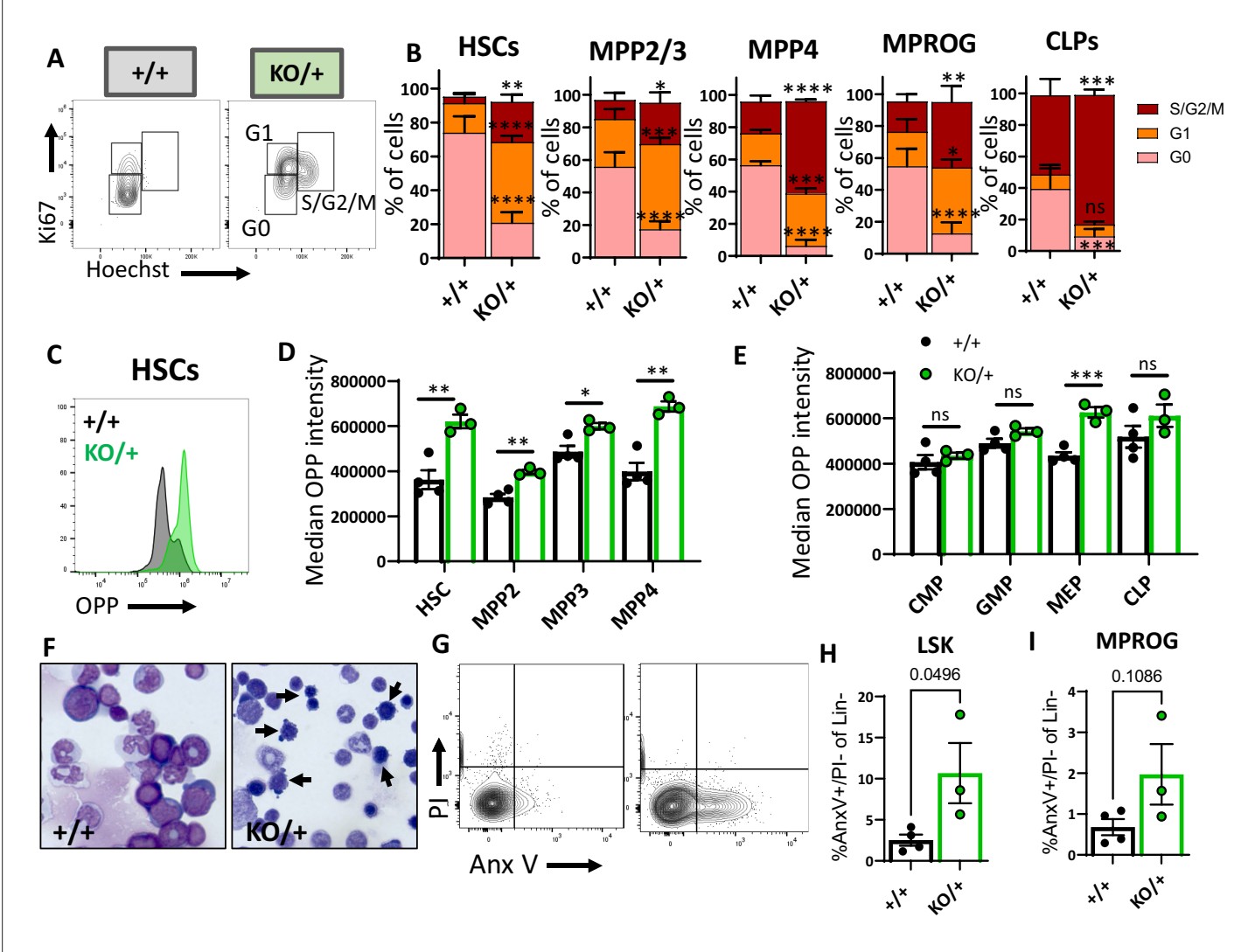

**Figure 6.** Hematopoietic stem cells (HSCs) and other hematopoietic progenitors have altered cycling, global translation levels, and apoptosis in *Rps12*^KO/+^ bone marrow. (**A**) Representative flow cytometry gating of HSCs (Flk2⁻CD48⁻LSK) cell cycle stages (G0, G1, S/G2/M) distribution determined by DNA (Hoechst) and Ki67 levels. (**B**) Cell cycle stages distribution in HSCs and in indicated progenitor populations. Asterisks correspond to p values assessing significant differences in each cell cycle stage between *Rps12*^KO/+^ and *Rps12*^+/+^ mice (6- to 8-week-old littermates, +/+ n=4, and KO/+ n=3). (**C**) Representative flow cytometry histogram showing OPP intensity in *RpS12*^KO/+^ (green) and *Rps12*^+/+^ (gray) HSCs. (**D, E**) Median o-propargyl puromycin (OPP) intensity of the indicated bone marrow populations (6- to 8-week-old littermates, +/+ n=4, and KO/+ n=3). This analysis was repeated in 6- to 7-month-old mice with similar results. (**F**) Representative images of bone marrow cytospins showing the high number of apoptotic cells (arrows) in *Rps12*^KO/+^ samples. (**G**) Representative flow cytometry gating of LIN- population showing apoptotic populations as determined by AnnexinV and PI staining. (**H, I**) Percentage of apoptotic (AnnexinV+) cells in LSK (Lin⁻cKit⁺Sca1⁺) and Myeloid progenitor (MPROG; Lin⁻cKit⁺Sca1⁻) populations (6- to 8-weeks-old littermates, +/+ n=4, and KO/+ n=3). Statistical analysis: quantifications represent mean +/− SEM, shown as the error bars, two-way ANOVA (**B–F**), and unpaired t-tests were performed to establish significance among populations between genotypes *p<0.05, **p<0.01, ***p<0.001, ****p<0.0001.

The online version of this article includes the following source data for figure 6:

**Source data 1.** Hematopoietic population cycling, apoptosis, and translation.

## Heterozygous post-natal loss of *Rps12* in the bone marrow impairs translation in HSCs and progenitors and leads to pancytopenia

To better understand the effects of *Rps12* deletion in HSCs, we generated a conditional knock-out model *Rps12*^flox/+^; Tal1-Cre-ERT, in which *Rps12* is excised specifically in hematopoietic cells upon tamoxifen (TAM) treatment. We induced *Rps12* deletion by tamoxifen injection in young adult mice

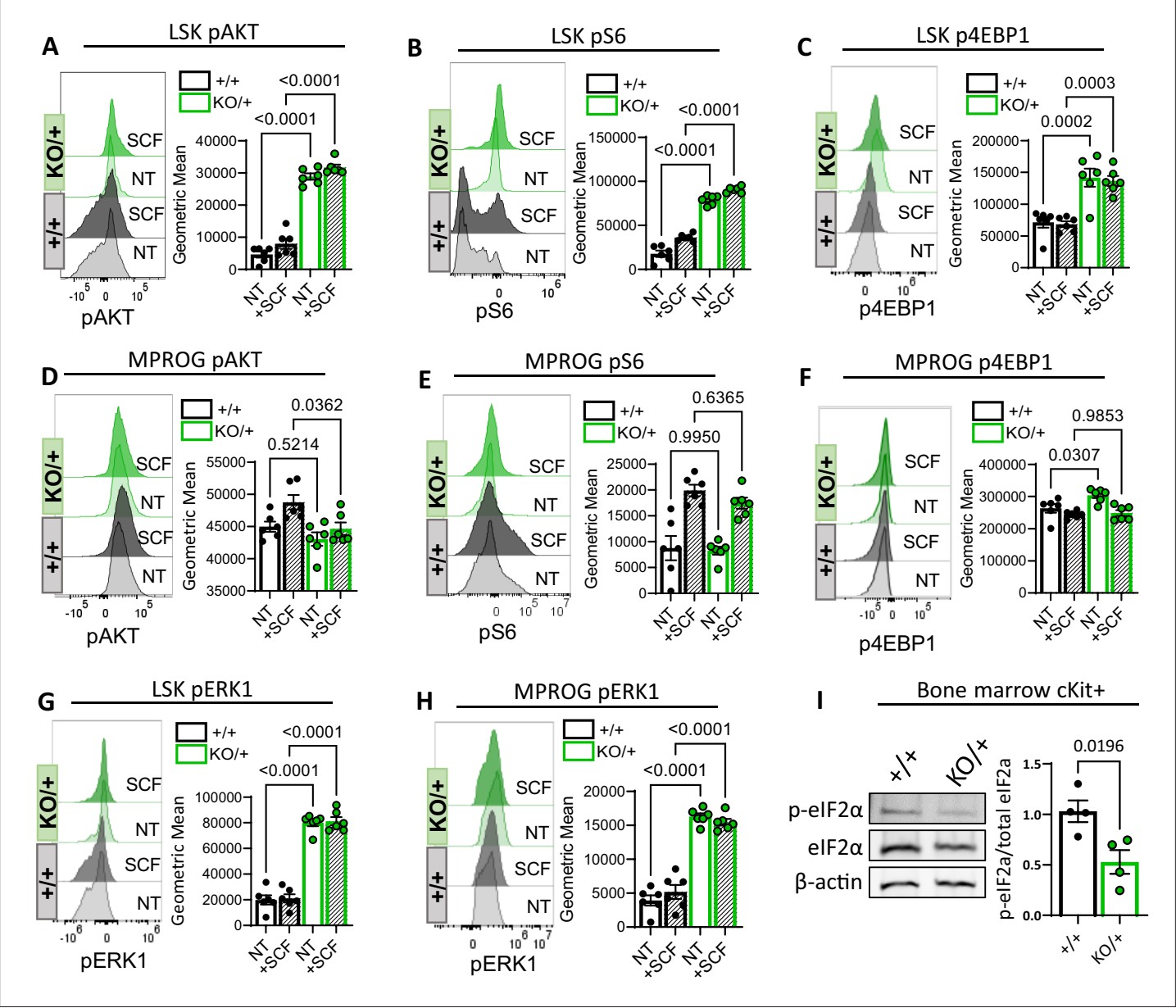

**Figure 7.** Decreased Rps12 levels lead to the excessive activation of the AKT/MTOR and ERK signaling pathways. (**A–H**) Representative phospho-flow cytometry histograms and quantification of the normalized geometric mean fluorescent intensity of pAKT (Ser 473) (**A, D**), pS6 (Ser235/236) (**B, E**), p4EBP1 (Thr37/46) (**C, F**), and pERK1(Thr202/Tyr204) (**D, H**) signal in the LSK (**A, B, C, G**) and MPROG (**D, E, F, H**) bone marrow cell populations. Baseline signal was determined in the none treated (NT) serum-starved cells, stimulation was done with the stem cell factor (SCF) ex vivo for 5 min. Immunophenotypic populations were defined as follows: <u>LSK:</u> Lin⁻cKit⁺Sca1⁺, <u>MPROG:</u> Lin⁻cKit⁺Sca1⁻. (**I**) Representative images of western blot analysis and quantification of phospho-eIF2α normalized to the total eIF2α protein in cKit-enriched BM samples (6- to 8-weeks-old littermates, +/+ n=4, and KO/+ n=4) (**A–H**) 7-weeks-old littermates, +/+ n=6, and KO/+ n=6 biological samples were used. Statistical analysis: quantifications represent mean +/− SEM, shown as the error bars, one-way ANOVA Tukey's multiple comparison tests were performed to establish significance among samples between genotypes.

The online version of this article includes the following source data for figure 7:

**Source data 1.** Blots of cKit + bone marrow cells.

**Source data 2.** pAKT, pS6, and pERK1 levels in hematopoietic progenitors.

and assessed changes in the peripheral blood and bone marrow at 7-, 12- and 14-weeks post TAM injection (*Figure 8A*). *Rps12* deletion by tamoxifen injection in young adult mice is present at 2- and 7-weeks post excision, as demonstrated by the presence of DNA fragment of 300 bp compared to the full WT length of 900 bp (*Figure 8—figure supplement 1A*). At 7 weeks post *Rps12* excision, there were no significant changes in peripheral blood counts (*Figure 8—figure supplement 1B*), in bone marrow cell counts, or by the absolute number of progenitor and stem cell populations in the bone marrow (*Figure 8—figure supplement 1C*). Additionally, there was no difference in translation levels in bone marrow HSPCs, measured using the OPP incorporation assay (*Figure 8—figure supplement 1D*). At 12 weeks post *Rps12* deletion, there were no changes in peripheral blood counts, total bone marrow cellularity, or in stem and progenitor cell populations in the bone marrow (*Figure 8B and C*). At that time point, however, OPP incorporation was significantly decreased in hematopoietic stem and progenitor cells, indicating decreased translation (*Figure 8D*). We did not observe any significant differences in the cell cycle profile of HSPCs at this time point (*Figure 8—figure supplement 1E*). At 14 weeks post *Rps12* deletion, we began to observe mild pancytopenia (*Figure 8E*), a significant decrease in bone marrow cellularity, a decrease in HSC numbers, and a trend toward decreased progenitor populations (*Figure 8F*). We did not observe any significant differences in OPP incorporation in HSPCs at this time point (*Figure 8—figure supplement 1F*). These data suggest that the effect of *Rps12* loss upon excision in adult hematopoietic cells is decreased translation, then decreases in HSCs maintenance and quiescence, resulting in decreased BM cellularity and pancytopenia.

## Discussion

We generated an *Rps12* knock-out mouse and describe the homozygous and heterozygous mutant phenotypes. Homozygous loss of *Rps12* was lethal during early embryogenesis. Heterozygous mice were viable with visible phenotypes and blood cell defects. Although the *Rps12* locus is unusual in *Drosophila*, in not exhibiting any haploinsufficient phenotype, and in playing a special role during cell competition, *Rps12* heterozygous mutant mice resembled mice mutants for other *Rp* genes in reduced body size, skeletal defects, and anemia. We also found that *Rps12* is required for erythroid differentiation. Some of the mice also exhibited hydrocephalus. Because aspects of the *Rps12* mutant phenotype resemble those of mice mutant for other *Rp* genes, including defective erythropoiesis, *Rps12* may also be a candidate gene for DBA (see below). Other Rps12 functions that have been suggested in mammalian cells and cancers could now be explored using this conditional knock-out model with tissue-specific Cre-drivers (*Derenzini et al., 2019*; *Brumwell et al., 2020*; *Katanaev et al., 2020*).

Most strikingly, we report that Rps12 is also crucial for normal hematopoietic stem cell maintenance, with defective engraftment and long-term repopulation of $Rps12^{KO/+}$ bone marrow in transplantation experiments. This seems specific to adult HSCs, as no hematopoietic defect was observed in the fetal livers of $Rps12^{KO/+}$ embryos. Although we have not determined whether the transition from the fetal liver to the bone marrow occurs normally, a defect then would only be expected to delay bone marrow engraftment, and does not seem sufficient to explain the chronically defective HSC function and striking loss of HSC quiescence observed in $Rps12^{KO/+}$ adult mice. The loss of bone marrow HSC quiescence was associated with chronic activation of Akt/MTOR and Erk, increased translation, and HSC apoptosis. Chronic activation of the AKT/MTOR pathway has been shown to result in increased HSC cycling, apoptosis, and decreased self-renewal (*Chen et al., 2008*; *Kharas et al., 2010*), which could explain the HSPC exhaustion phenotype and increased HSPC apoptosis in $Rps12^{KO/+}$ mice.

The increase in HSPC-specific global translation upon heterozygous embryonic deletion of *Rps12* is the opposite of what has been reported in $Rpl24^{Bst/+}$ and $Rps14^{+/-}$ (*Signer et al., 2014*; *Schneider et al., 2016*), although there are other *Rp* genotypes where HSC cycling is increased (*Terzian et al., 2011*; *Schneider et al., 2016*). However, it is consistent with the increased translation that has been reported in mice with deletion of *Pten* in HSCs, which leads to activation of the AKT/MTOR pathway (*Signer et al., 2014*). Such an increase would in fact be expected as a result of the striking activation of the AKT/MTOR and ERK pathways. $Rps12^{KO/+}$ c-Kit+ hematopoietic progenitors also have lower phosphorylated eIF2α, which would also predict higher translation levels. Importantly, however, detailed studies of *Rps12* depletion from ribosomes in yeast clearly demonstrate a strong reduction in translation (*Martín-Villanueva et al., 2020*).

Inducible, conditional post-natal deletion of *Rps12* in hematopoietic cells revealed a sequence of events over time (summarized in (*Figure 8G*)). Loss of *Rps12* in adult hematopoietic cells led to

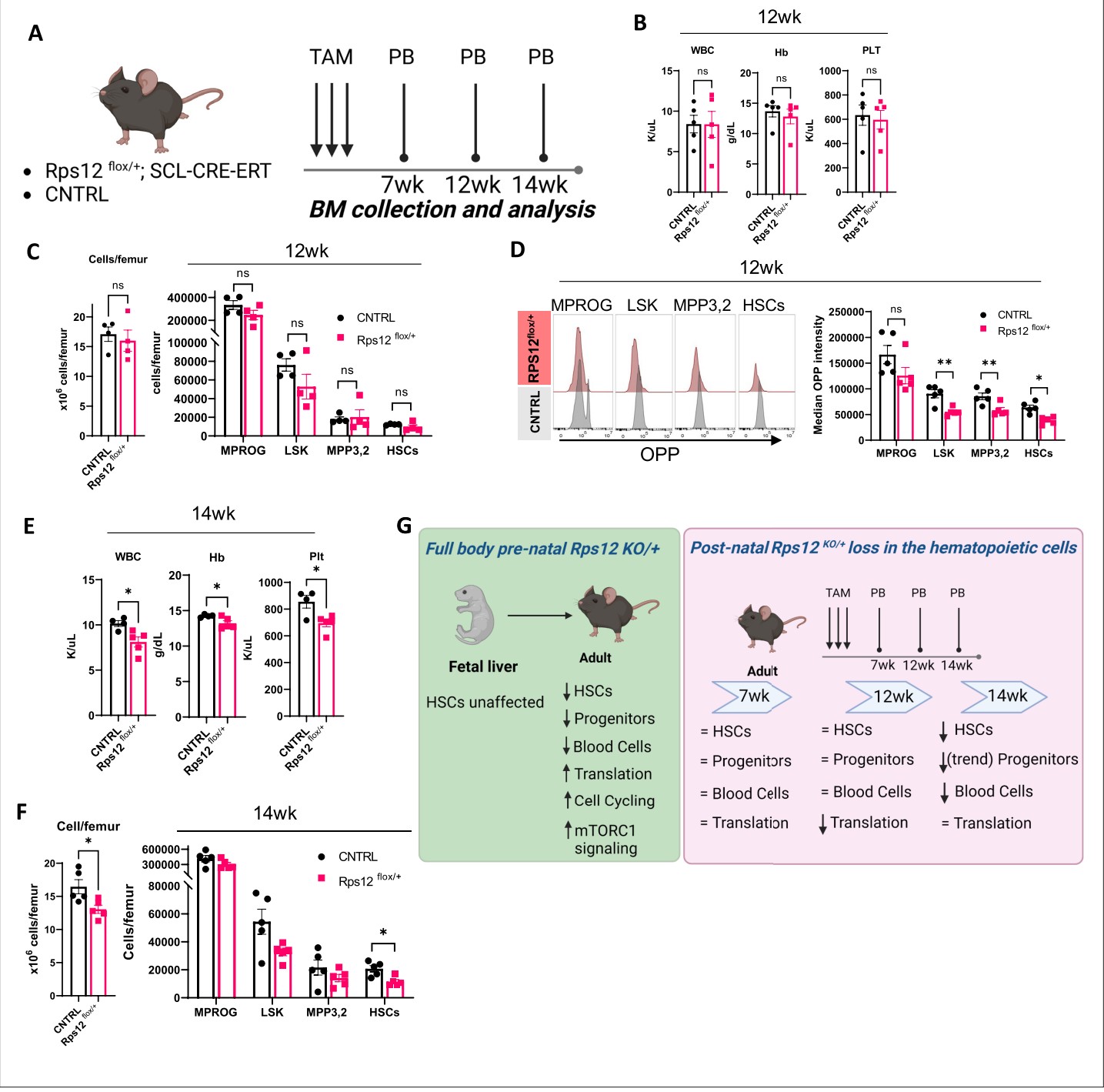

**Figure 8.** Heterozygous loss of *Rps12* in the hematopoietic cells impairs translation in the hematopoietic stem and progenitor cells and leads to pancytopenia. (**A**) Schematic representation of the time-course analysis testing the effect of the heterozygous loss of *Rps12* activity in hematopoietic stem cells (HSCs). (**B–D**) Analysis of *Rps12*flox/+; Tal1-Cre-ERT (*Rps12*flox/+) and *Rps12*flox/+(CNTRL) at 12 weeks post tamoxifen treatment (TAM): (**B**) Peripheral blood cell analysis, (**C**) quantification of bone marrow cellularity and absolute cell count of bone marrow populations per femur, (**D**) representative flow histograms, and quantification of the median OPP intensity in bone marrow populations. (**E**) Peripheral blood cell analysis at 14 weeks and (**F**) quantification of bone marrow cellularity and absolute cell count of bone marrow populations per femur. (**G**) Schematic summary of the phenotypic analysis of mouse models. Immunophenotypic populations were defined as follows: MPROG: Lin⁻cKit⁺Sca1⁻, LSK: Lin⁻cKit⁺Sca1⁺; MPP3,2: Lin⁻cKit⁺Sca1⁺Flk2⁻CD48⁺, HSCs: Lin⁻cKit⁺Sca1⁺Flk2⁻CD48⁻. Statistical analysis: quantifications represent mean +/− SEM, shown as the error bars. Significance was determined using the Welch's t-test *p≤0.05, **p≤0.01. At each time point n (*Rps12*flox/+)=5 and n (CNTRL)=5. Schematics created with BioRender.

*Figure 8 continued on next page*

*Figure 8 continued*

The online version of this article includes the following source data and figure supplement(s) for figure 8:

**Source data 1.** Conditional KO hematopoietic populations.

**Figure supplement 1.** Heterozygous loss of *Rps12* in the hematopoietic cells.

**Figure supplement 1—source data 1.** Analysis of Rps12 SCL-CRE excised hematopoietic progenitors.

decreased translation in HSPCs, and then a decreasing HSC pool over time, with decreased BM cellularity and pancytopenia, but no significant effects on HSC quiescence. Thus, it appears that initially Rps12 is required to maintain translation levels in the adult hematopoietic system. The more complicated phenotype of the full-body *Rps12*^KO/+^ mouse may reflect an earlier developmental effect on the BM, and a potential effect on the BM microenvironment. Furthermore, this phenotype could reflect compensatory changes due to loss of HSC maintenance, reflected in the observed increases in MTOR signaling, translation, and HSC cycling.

The fact that *Rps12*^KO/+^ mice exhibit fully-penetrant pancytopenia with a severe bone marrow failure phenotype, in addition to the erythropoiesis defect, raises the possibility that *Rps12* might not have been found mutated in DBA patients due to a more severe human phenotype of *Rps12* mutation that is not classified as DBA. Perhaps only a hypomorphic *Rps12* genotype would be associated with DBA. While caution is required extrapolating from mouse phenotypes to humans, our study nevertheless raises the possibility that *Rps12* could in fact be a candidate gene not only for DBA, but for a broader group of bone marrow failure disorders. One reason that genetic alterations in *Rps12* have not yet been reported in DBA or other bone marrow failure disorders could be that *Rps12* is not included in the most common targeted next-generation sequencing (NGS) panels used in the diagnosis of these disorders, some of which report molecular diagnostic rates of only 44–59% (*Ghemlas et al., 2015*; *Muramatsu et al., 2017*; *Gálvez et al., 2021*). We suggest that *Rps12* should be included in expanded NGS panels for bone marrow failure disorders or should be sequenced in patients in whom there are no molecular findings in the standard NGS panels.

## Methods

**Key resources table**

| Reagent type (species) or resource | Designation | Source or reference | Identifiers | Additional information |
|---|---|---|---|---|
| Genetic reagent (*Mus musculus*) | C57BL/6 J | The Jackson Laboratory | RRID:IMSR_JAX:000664 | |
| Genetic reagent (*Mus musculus*) | EIIa-Cre (FVB/N-Tg(EIIa-cre)C5379Lmgd/J) | The Jackson Laboratory | RRID:IMSR_JAX:003724 | |
| Genetic reagent (*Mus musculus*) | B6.SJL-Ptprca/BoyAiTac (CD45.1) | The Jackson Laboratory | RRID:IMSR_JAX:002014 | |
| Genetic reagent (*Mus musculus*) | RpS12^flox/flox^ | This study | Rps12^em1Nbakr^ MGI:6388411 | Generated for this study |
| Genetic reagent (*Mus musculus*) | Scl-Cre-ER | The Jackson Laboratory | RRID:IMSR_JAX:037466 | |
| Genetic reagent (*Mus musculus*) | RpS12^flox/+;^ Scl-Cre-ER | This study | | Generated for this study |
| Antibody | Rat monoclonal CD16/CD32 | BD biosciences | Cat # 553142 | Clone: 2.4G2 (1 µl per $10^7$ cells) |
| Antibody | APC-Cy7 Rat anti-mouse monoclonal Gr1 | BD biosciences | Cat # 557661 | Clone: RB6-8C5 (1 µl per $10^7$ cells) |
| Antibody | PE rat anti-mouse monoclonal Mac1 | BioLegend | Cat # 101208 | Clone: M1/70 (1 µl per $10^7$ cells) |
| Antibody | Alexa Fluor 700 Rat anti-Mouse monoclonal B220 | BD biosciences | Cat # 557957 | Clone: RA3-6B2 (1 µl per $10^7$ cells) |
| Antibody | APC Rat Anti-Mouse monoclonal Thy1.2 | BD biosciences | Cat # 553007 | Clone: 53–2.1 (1 µl per $10^7$ cells) |

*Continued on next page*

*Continued*

| Reagent type (species) or resource | Designation | Source or reference | Identifiers | Additional information |
| --- | --- | --- | --- | --- |
| Antibody | FITC Mouse Anti-Mouse monoclonal CD45.1 | BD biosciences | Cat # 553775 | Clone: A20 (1 µl per $10^7$ cells) |
| Antibody | PE/Dazzle 594 mouse anti-mouse monoclonal CD45.2 | BioLegend | Cat # 109845 | Clone:104 (1 µl per $10^7$ cells) |
| Antibody | CD3e hamster monoclonal, Biotin, | eBioscience | Cat # 13003182 | Clone: 145–2 C11 (1 µl per $10^7$ cells) |
| Antibody | CD4 rat Monoclonal, Biotin, | eBioscience | Cat #13004182 | Clone: GK1.5 (1 µl per $10^7$ cells) |
| Antibody | CD8a rat Monoclonal, Biotin, | eBioscience | Cat # 13008182 | Clone: 53–6.7 (1 µl per $10^7$ cells) |
| Antibody | Gr1 rat Monoclonal, Biotin, | eBioscience | Cat # 13593182 | Clone: RB68C5 (1 µl per $10^7$ cells) |
| Antibody | B220 rat Monoclonal, Biotin, | eBioscience | Cat # 13045282 | Clone: RA3-6B2 (1 µl per $10^7$ cells) |
| Antibody | CD19 mouse Monoclonal, Biotin, | eBioscience | Cat # 13019182 | Clone: MB 19–1 (1 µl per $10^7$ cells) |
| Antibody | Ter119 rat Monoclonal, Biotin, | eBioscience | Cat # 13592182 | Clone: TER119 (1 µl per $10^7$ cells) |
| Antibody | PE/Cyanine7 rat anti-mouse monoclonal Sca-1 | BioLegend | Cat #108114 | Clone: D7 (1 µl per $10^7$ cells) |
| Antibody | APC Rat Anti-Mouse monoclonal cKit | BD Biosciences | Cat #553356 | Clone:2B8 (1 µl per $10^7$ cells) |
| Antibody | PE Rat Anti-Mouse monoclonal FCRg | Thermo Fisher | Cat # 12-0161-82 | Clone: 93 (1 µl per $10^7$ cells) |
| Antibody | eFluor 450 Rat Anti-Mouse monoclonal CD34 | Thermo Fisher | Cat #48-0341-80 | Clone:RAM34 (1 µl per $10^7$ cells) |
| Antibody | Brilliant Violet 421 Rat anti-Mouse monoclonal CD150 | BD Biosciences | Cat #562811 | Clone: Q38-480 (1 µl per $10^7$ cells) |
| Antibody | Alexa Fluor 700 hamster anti-mouse monoclonal CD48 | BioLegend | Cat #103426 | Clone: HM48-1 (1 µl per $10^7$ cells) |
| Antibody | PE/Cyanine5 Rat anti-mouse monoclonal Flk-2 | BioLegend | Cat #135312 | Clone: A2F10 (1 µl per $10^7$ cells) |
| Antibody | Brilliant Violet 605 mouse anti-mouse monoclonal CD45.2 | BioLegend | Cat #109841 | Clone: 104 (1 µl per $10^7$ cells) |
| Antibody | PerCP/Cyanine5.5 rat anti-mouse monoclonal IL-7Rα | BioLegend | Cat #135021 | Clone: A7R34 (1 µl per $10^7$ cells) |
| Other | Streptavidin-APC-Cy7 | BD Biosciences | Cat # 554063 | second-step reagent for cells stained with biotinylated primary antibodies |
| Antibody | PE anti-mouse monoclonal Flk-2 | BioLegend | Cat # 135306 | Clone: A2F10 (1 µl per $10^7$ cells) |
| Commercial assay, kit | FITC Annexin V | BD Biosciences | Cat # 560931 | Apoptosis detection: Phosphatidylserine probe |
| Commercial assay, kit | Propidium Iodide | BD Biosciences | Cat # 556463 | Apoptosis detection: Staining Solution |
| Antibody | pS6 (Ser235/236) Alexa 488 rabbit monoclonal | Cell Signaling | Cat # 4803 S | Clone: D57.2.2E ICFC (1:50) |
| Antibody | pERK1 (T202/Y204) Alexa 488 mouse monoclonal | Cell Signaling | Cat # 4374 | Clone: E10 ICFC (1:20) |
| Antibody | pAkt (Ser473) AF647 rabbit monoclonal | Cell Signaling | Cat # 2337 | Clone: 193H12 ICFC (1:20) |

*Continued on next page*

*Continued*

| Reagent type (species) or resource | Designation | Source or reference | Identifiers | Additional information |
|---|---|---|---|---|
| Antibody | p4E-BP1 (Thr37/46) rabbit monoclonal AF647 | Cell Signaling | Cat # 5123 | Clone: 236B4<br>ICFC (1:20) |
| Antibody | FITC Mouse monoclonal Anti-Ki-67 | BD Biosciences | Cat # 556026<br>RRID: AB_396302 | Clone: B56<br>(1 µl per $10^7$ cells) |
| other | Hoechst 33342 Solution | BD Biosciences | Cat # 561908 | See legend to *Figure 6* |
| Commercial assay, kit | Click-iT Plus OPP Alexa Fluor 488 | Thermo Fisher | Cat # C10456 | Protein Synthesis Assay Kit |
| Antibody | APC rat anti-mouse monoclonal TER-119 | BioLegend | Cat # 116211 | Clone: TER119<br>(1 µl per $10^7$ cells) |
| Antibody | PE rat anti-Mouse monoclonal CD71 | BD biosciences | Cat # 561937 | Clone: C2<br>(1 µl per $10^7$ cells) |
| Sequence-based reagent | gRNA 1 | This study | PCR primer | CGCAGTAGACACG<br>CTATCGCCGG |
| Sequence-based reagent | gRNA 2 | This study | PCR primer | GTGGGTTGCTGT<br>GTGGATCGGGG |
| Sequence-based reagent | F1 | This study | PCR primer | GCACATGCGC<br>ACAGAAGT |
| Sequence-based reagent | R1 | This study | PCR primer | CGGACTATCTA<br>TCCCCACGA |
| Sequence-based reagent | F2 | This study | PCR primer | GTACAGCTATC<br>TGCCAGGAA |
| Sequence-based reagent | R2 | This study | PCR primer | CGAGGTCGACGGTATCG |
| Sequence-based reagent | F3 | This study | PCR primer | CGATACCGTCGACCTCG |
| Sequence-based reagent | R3 | This study | PCR primer | GTGCTAGCAAC<br>AGAAGGTTC |
| Sequence-based reagent | F4 | This study | PCR primer | GTCTCAATACTGTGGGGTGT |
| Commercial assay, kit | Cytofix/Cytoperm | BD biosciences | Cat # BDB554714 | Fixation/Permeabilization Solution Kit |
| Commercial assay, kit | DNeasy kit | Qiagen | Cat #69504 | DNA extraction kit |
| Chemical compound, drug | mSCF | Peprotech | Cat # 250–03 | Cytokine |
| Other | MethoCult GF M3434 | Stem Cell Technologies | Cat # 03434 | Methylcellulose media |
| Other | MethoCult M3334 | Stem Cell Technologies | Cat # 03334 | Methylcellulose media |
| Other | CD117 MicroBeads, mouse | Miltenyi Biotec | Cat #130-091-224 | See 'Western blot analysis' in methods section |
| Other | LS Column | Miltenyi Biotec | Cat # 130-042-401 | Separation column |
| Antibody | RpS12, rabbit polyclonal | Proteintech | Cat # 16490–1-AP | 1:1000 |
| Antibody | β-Actin (13E5) Rabbit monoclonal mAb | Cell Signaling | Cat # 4970 | 1:1000 |
| Antibody | eIF2α (D7D3) Rabbit monoclonal mAb | Cell Signaling | Cat # 5324 | 1:1000 |
| Antibody | phospho-eIF2α (Ser52) rabbit polyclonal | Thermo Scientific | Cat # BS-4842R | 1:1000 |
| Commercial assay, kit | miRNA First-Strand Synthesis Kit | Takara Bio | Cat # 638313 | |
| Commercial assay, kit | mirVana miRNA Isolation Kit | Thermo Fisher | Cat # AM1560 | |

| Reagent type (species) or resource | Designation | Source or reference | Identifiers | Additional information |
|---|---|---|---|---|
| Other | Power SYBR Green Master Mix | Applied Biosystems | Cat # 4367659 | PCR reaction mix |

## Mice

All animals were housed at the Animal Housing and Studies Facility at Albert Einstein College of Medicine (AECOM) under pathogen-free conditions and experiments were performed following protocols approved by the Institutional Animal Care and Use Committee (IACUC) (Protocol #20181206). C57BL/6 J and EIIa-Cre (FVB/N-Tg(EIIa-Cre)C5379Lmgd/J) mice were obtained from Jackson. B6.SJL-Ptprca/BoyAiTac (CD45.1) mice from Taconic were used for transplantation experiments. To generate *Rps12*$^{KO/+}$ mice, we crossed *Rps12*$^{flox/flox}$ to EIIa-Cre mice and used primers flanking the floxed region to identify progeny where recombination had occurred. Unless indicated, these mice were kept as heterozygous by crossing them with C57BL/6 J, and the presence of the EIIa-Cre transgene was crossed out. In each case, the genotypes were confirmed by PCR using genomic DNA extracted from tails using a DNeasy kit from Qiagen (#69504). Peripheral blood samples were collected via facial vein bleeding under isoflurane anesthesia, and blood counts were obtained using the Genesis analyzer (Oxford Science). To generate growth curves, 5-day-old pups were genotyped and numbered by cutting toes. Pups' weight was measured daily from day 5–21 of age.

## Generation of *Rps12*$^{flox}$ knock-in mice

A pair of guide-RNAs (gRNAs) targeting intron 1 and intron 3 of the *Rps12* gene, respectively were designed by an online tool (http://crispr.mit.edu/) and generated by in vitro transcription. Cas9 mRNA was purchased from SBI. An *Rps12* conditional knockout homology-directed repair (HDR) plasmid containing 2 kb homologous arms at each side and exons 2 and 3 flanked by loxP sites (*Figure 1—figure supplement 1*) was generated by SLiCE cloning. Super ovulated female C57BL/6 J mice (3–4 weeks old) were mated to C57BL/6 J males, and fertilized embryos were collected from oviducts. The gRNAs, Cas9 mRNA and conditional knockout HDR plasmid were microinjected into the cytoplasm of fertilized eggs. The injected zygotes were transferred into pseudo-pregnant CD1 females and the resulting pups were genotyped. Out of 20 pups, two mice were identified as *Rps12*$^{flox/+}$, which were then crossed to obtain *Rps12*$^{flox/flox}$ (*Rps12*$^{em1Nbakr}$ MGI:6388411). The corresponding DNA sequences can be found in the Key Resources Table.

## Generation of the *Rps12*$^{KO}$ knock-out mutation

*Rps12*$^{flox/flox}$ mice were mated to EIIA-Cre mice for excision during embryogenesis, the F1 generation carrying the potential *Rps12* excision bred with C57BL/6 J mice, and the F2 generation genotyped to identify *Rps12*$^{KO/+}$ mice lacking EIIA-Cre.

## Generation of *Rps12*$^{flox/+}$; Tal1-Cre-ERT mice

*Rps12*$^{flox/flox}$ mice were mated to the Tal1-ERT-Cre mice and F1 generation of *Rps12*$^{flox/+}$; Tal1-Cre-ERT mice. 6–8 weeks old animals were treated with three doses of tamoxifen (3 × 4 mg/mouse) to achieve the *Rps12* excision.

## Preparation of single-cell suspensions

Bone marrow single-cell suspension was prepared from freshly harvested femurs, tibiae, ilia, and vertebrae by gentle crushing of the bones in phosphate-buffered saline containing 2% fetal bovine serum (PBS/2% FBS) followed by filtration through a 40 μm strainer. Spleen cells were obtained from freshly harvested spleens. Single-cell suspension was prepared by dissociating spleens using the flat end of a plunger against 40 μm strainers and washed with PBS/2% FBS. Fetal liver single cell suspension was prepared from freshly harvested E13.5 fetal livers by passing through a 200 μl pipet tip and filtered through a 40 μm strainer in PBS/2% FBS. Cells were subjected to RBC lysis (Qiagen) according to the manufacturer's protocol and used for the further steps described below. To calculate the absolute number of cells per femur one femur per mouse was flushed, RBC lysed and cells counted to obtain the total number of cells per femur.

## Flow cytometry on live cells

### Bone marrow

Single-cell bone marrow suspensions were stained with a cocktail of biotin-conjugated lineage antibodies for 30 min at 4 °C, washed with PBS/2% FBS, stained with fluorochrome-conjugated antibody cocktails for 30 min at 4 °C, washed with PBS/2% FBS, resuspended in PBS/2% FBS, filtered through a 40 µm strainer, and subjected to Flow analysis. For the antibody panels, refer to *Supplementary file 1*.

### Peripheral blood

samples were subjected to RBC lysis, blocked with CD16/CD32 10 min at 4 °C followed by staining with fluorochrome-conjugated antibodies, washed with PBS/2% FBS, resuspended in PBS/2% FBS, filtered through a 40 µm strainer and subjected to Flow analysis. For the antibody panels, refer to *Supplementary file 1*.

### Erythropoiesis analysis

Obtained single cell suspension of spleen cells (without RBC lysis) was blocked with CD16/CD32 for 10 min at 4 °C and stained with fluorochrome-conjugated antibodies for 30 min at 4 °C (*Supplementary file 1*). E13.5 fetal livers single cell suspension was blocked with CD16/CD32 10 min at 4 °C and stained with erythropoiesis or progenitor panels as described above. All blocking and staining steps were performed in PBS/2% FBS.

### Apoptosis analysis

Single-cell suspension samples were stained with the lineage-cocktail and fluorochrome-conjugated antibody cocktails as described above. After completion of surface antibody staining, samples were incubated with FITC-conjugated Annexin V (BD-560931) and Propidium Iodide (BD-556463) following the manufacturer's instructions.

All flow cytometry was performed with BD FACS LSRII or Cytek Aurora and data analysis was done with FlowJo Software (v9, v10).

## Flow cytometry on fixed cells

### Cell cycle

Fresh single-cell suspension of the RBC lysed bone marrow cells was stained with lineage antibodies followed by staining with fluorochrome-conjugated antibodies against surface markers, as described above. Immediately after staining cells were fixed and permeabilized using Cytofix/Cytoperm Fixation/Permeabilization Solution Kit (BD biosciences, BDB554714) according to the manufacturer's instructions. Following fixation, cells were incubated overnight at 4 °C with FITC-conjugated Ki67 antibody in Perm/Wash buffer. DNA was stained with 25 µl/ml Hoechst in Perm/Wash buffer before flow cytometry analysis. Flow cytometry was performed with BD FACS LSRII or Cytek Aurora and data analysis was done with FlowJo Software (v9, v10).

### Global translation in vitro

protocol was based on a previously described assay (*Signer et al., 2014*). Single-cell bone marrow RBC lysed cell suspension was obtained as described. Cells were resuspended in DMEM (Corning 10–013-CV) media supplemented with 50 µM β-mercaptoethanol (Sigma) and 20 µM OPP (Thermo Scientific C10456). Cells were incubated for 45 min at 37 °C and then washed with $Ca^{2+}$ and $Mg^{2+}$ free PBS. The samples were stained with biotin-labeled antibodies, followed by staining with fluorochrome-conjugated antibody cocktails, fixed and permeabilized using Cytofix/Cytoperm as described above. After permeabilization with Perm/Wash buffer, cells were resuspended in Click-iT Plus Reaction Cocktail (Thermo Scientific C10456) containing azide conjugated to Alexa Fluor 488 for 30 min at room temperature, washed once with Click-iT Reaction Rinse Buffer and resuspended in Perm/Wash buffer. Flow cytometry was performed with Cytek Aurora and data analysis was done with FlowJo Software (v9, v10).

### Phospho-flow cytometry

Bone marrow cells were starved for 1 hr in IMDM 2% FBS at 37 °C, stained with lineage antibodies, followed by staining with fluorochrome-conjugated antibodies against surface markers, as described

above. Post-staining cells were stimulated with 100 ng/ml mSCF (Peprotech #250–03) in 2% PBS-FBS for 5 min at 37 °C. Stained and stimulated cells were fixed and permeabilized with Cytofix/Cytoperm as described above and stained with phospho-S6 (Ser235/236) - Alexa 488 (Cell Signaling Technology, 4803 S) (1:100) and phospho-AKT (Ser473) - Alexa647 (Cell Signaling Technology, 2337 S), phospho-4E-BP1 (Thr37/46) - Alexa Fluor647 -(Cell Signaling Technology, 5123 S) of pERK1(T202/Y204) - Alexa 488 (Cell Signaling 4374) at 1:20 dilutions. Cells were washed with Perm/Wash buffer to remove residual and unbound antibodies, and resuspended in fresh Perm/Wash buffer, followed by flow cytometry analysis on the Cytek Aurora. Analysis of all flow cytometry data was performed using FlowJo software (v9, v10).

## Methylcellulose cultures and serial re-plating

Single-cell bone marrow suspensions (post RBC lysis) or single-cell fetal liver cell suspensions (without RBC lysis) were resuspended in RPMI media supplemented with 10% FBS and 1% penicillin/streptomycin. Cells were manually counted on a hemocytometer using Trypan blue and plated in methylcellulose media (M3434 or M3334, Stem Cell Technologies) at a density of $5 \times 10^5$ live cells/ml (in M3434) or $10^4$ live cells/ml (in M3334) in 35 mm cell culture plates. Samples were incubated at 37 °C in 6.5% $CO_2$ at constant humidity. Colonies were scored and evaluated 7–10 days after plating. To replate, cells were washed from the plates with RMPI media, counted, and re-plated in fresh M3334 methylcellulose at a density of $10^4$ live cells/dish in 35 mm plates. This process was repeated until cell exhaustion in one of the experimental groups.

## Bone marrow transplantation

6- to 8-weeks-old B6.SJL (CD45.1) recipient mice were lethally irradiated with a single dose of 950 Gy using a Cesium-137 gamma-ray irradiator (Mark I irradiator Model 68) at least 3 hr before transplantation. For non-competitive assays, $10^6$ whole bone marrow cells from a donor control ($Rps12^{flox/flox}$ or $Rps12^{flox/+}$) or $Rps12^{KO/+}$ mouse (CD45.2) were injected into the retro-orbital venous sinus of recipient mice under isoflurane anesthesia. For competitive transplants, $10^5$ whole bone marrow cells from control $Rps12^{flox/+}$ or $Rps12^{KO/+}$ donor mouse (CD45.2), and $10^5$ competitor cells from a B6.SJL (CD45.1) mouse were injected into each recipient mouse. Mice were given drinking water treated with 100 mg/ml Baytril100 (Bayer) for 3 weeks after transplantation. Peripheral blood was collected every 4 weeks and animals were euthanized at the specified experimental time points.

## Western blot analysis

Whole bone marrow cells were enriched for cKit + cells using CD117 MicroBeads and MACS LS Columns (Miltenyi Biotec 130-091-224 and 130-042-401) following the manufacturer's protocol. $2 \times 10^6$ cKit enriched cells were resuspended in 150 µl Laemmli buffer (BioRad 1610737) supplemented with 1:10 β-mercaptoethanol, passed through a 25 G needle to break the DNA and incubated at 95 °C for 5 min. An equal amount of each sample was separated in polyacrylamide gels (BioRad 4568081), transferred to nitrocellulose membrane (Licor), and blocked (Licor 927–90001). IRDye near-infrared secondary antibodies (Licor) were used to visualize the proteins. The following primary antibodies were used: RpS12 (Proteintech; polyclonal), β-actin (Cell Signaling; 13E5), eIF2α (Cell signaling, 5324T), phospho-eIF2α (Thermo Scientific; Ser52; polyclonal).

## RNA extraction, RT-qPCR, and bioanalyzer analysis

Total RNA was extracted from $2 \times 10^6$ cKit enriched cells (isolated using CD117 MicroBeads as previously described) with mirVana miRNA Isolation Kit (Thermo Fisher AM1560) following the manufacturer's instructions for the isolation of total RNA. Final RNA concentration was determined using a NanoDrop Mircrovolume Spectrophotometer (Thermo Scientific). For RT-qPCR, cDNA synthesis was performed using 405 ng of total RNA and the Mir-X miRNA First-Strand Synthesis Kit (Takara Bio 638313) as recommended by the manufacturer. cDNA samples were analyzed, in duplicates, using Power SYBR Green Master Mix (Applied Biosystems 4367659) in an Applied Biosystems StepOne Real-Time PCR instrument (Applied Biosystems 4376357). For mRNA analysis, GAPDH, HPRT1, and RER1 mRNA levels were used as internal controls, and for snoRNA analysis, *U6*, *Sno202*, and *Sno234*, small RNAs were used as internal controls.

## Histology

Peripheral blood smears and cytospins from RBC lysed bone marrow samples were stained using the Hema 3 System (Fisher) following the manufacturer's instructions. The images were acquired using a Zeiss Axiovert microscope with a digital camera.

## Statistical methods

Two-tailed Student's t-tests were performed to compare the statistical significance between the two samples. When comparing more than two groups, one-way ANOVA tests were performed with the Turkey's multiple comparison test. For the presence/absence of phenotype, statistical significance was calculated with Fisher's exact test. Analysis was done using GraphPad Prism v9.

## Reagents

All antibodies used for flow cytometry assays, and primers used for CRISPR gene editing and PCR can be found in *Supplementary file 1* and the Key Resources Table. Other reagents are mentioned in the methods section and Key Resources Table.

## Acknowledgements

This work was supported by National Institutes of Health grants R01GM104213 (to NEB), R01CA196973 (to KG), R56DK130895 (to KG), R01DK130895 (to KG), an award from the Albert Einstein College of Medicine Human Genetics Program (to NEB), startup funds from the Albert Einstein College of Medicine and Albert Einstein Cancer Center (to KG), the NHLBI/NIH Ruth L Kirschstein National Research Service Award F32HL146119 (to KA), and the IRACDA/BETTR training Institutional Research and Academic Career Development Award 2K12GH102779-07A1 (to KA). For flow cytometry this work utilized the analyzers Cytek Aurora Multiparameter Flow Cytometer and BD LSR-II with the help of Dr. Jinghang Zhang, Dr. Yu Zhang, and Aodengtuya Fnu. The Cytek Aurora Multiparameter Flow Cytometer was purchased with funding from the National Institutes of Health SIG grant #1S10OD026833-01. Data in this paper are from a thesis to be submitted in partial fulfillment of the requirements for the Degree of Doctor of Philosophy in the Biomedical Sciences, Albert Einstein College of Medicine.

## Additional information

### Competing interests

Kira Gritsman: has received research funding from iOnctura, SA, and ADC Therapeutics. The other authors declare that no competing interests exist.

### Funding

| Funder | Grant reference number | Author |
|---|---|---|
| National Institute of General Medical Sciences | R01GM104213 | Nicholas E Baker |
| National Institute of Diabetes and Digestive and Kidney Diseases | R56DK130895 | Kira Gritsman |
| National Institute of Diabetes and Digestive and Kidney Diseases | R01DK130895 | Kira Gritsman |
| National Institutes of Health | F32HL146119 | Kristina Ames |
| National Institutes of Health | 2K12GH102779 | Kristina Ames |
| Albert Einstein College of Medicine | Human Genetics Program | Kira Gritsman |

| Funder | Grant reference number | Author |
|---|---|---|
| National Institutes of Health | R01CA196973 | Kira Gritsman |
| National Institute of General Medical Sciences | IRACDA/BETTR training Institutional Research and Academic Career Development Award 2K12GH102779-07A1 | Kristina Ames |

The funders had no role in study design, data collection and interpretation, or the decision to submit the work for publication.

## Author contributions

Virginia Folgado-Marco, Kristina Ames, Conceptualization, Data curation, Formal analysis, Validation, Investigation, Visualization, Methodology, Writing – original draft; Jacky Chuen, Investigation; Kira Gritsman, Conceptualization, Formal analysis, Supervision, Funding acquisition, Visualization, Project administration, Writing – review and editing; Nicholas E Baker, Conceptualization, Supervision, Funding acquisition, Project administration, Writing – review and editing

## Author ORCIDs

Kristina Ames  http://orcid.org/0000-0002-6845-0898
Kira Gritsman  http://orcid.org/0000-0002-1367-1167
Nicholas E Baker  http://orcid.org/0000-0002-4250-3488

## Ethics

This study was performed in strict accordance with the recommendations in the Guide for the Care and Use of Laboratory Animals of the National Institutes of Health. All of the animals were handled according to approved Institutional Animal Care and Use Committee (IACUC) protocols of the Albert Einstein College of Medicine (Protocol #20181206). All procedures were performed under isoflurane anesthesia to minimize animal suffering.

## Decision letter and Author response

Decision letter https://doi.org/10.7554/eLife.69322.sa1
Author response https://doi.org/10.7554/eLife.69322.sa2

# Additional files

## Supplementary files

• Supplementary file 1. Antibodies used for flow cytometry panels of peripheral blood and bone marrow samples. This file contains a list of all the biotinylated and fluorochrome-conjugated antibodies used in both live and fixed cell samples used in the experiments described in this paper. The antibodies are grouped by flow cytometry panel and include clone name as well as the manufacturer and catalog number information for ease of use and reproducibility. Additional information on antibody concentration used for staining can be found in the 'Key Resources Table.'

• Transparent reporting form

## Data availability

There are no large-scale datasets associated with this paper. All data generated or analyzed during this study are included in the manuscript and supporting files.

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
