## [Editor Report]

This paper shows that haploinsufficiency of the ribosomal protein gene Rps12 in mice results in a number of phenotypes including defects in erythropoiesis, chronic pancytopenia, and loss of hematopoietic stem cell quiescence. This work will significantly add to the growing body of evidence that specific cell populations are particularly sensitive to global changes in mRNA translation. The manuscript contributes to our understanding of ribosome formation and function, mRNA translation, development, and stem cell biology.

---

## [Decision Letter]

**Decision letter after peer review:**

Thank you for submitting your article "Haploinsufficiency of the essential gene RpS12 causes defects in erythropoiesis and hematopoietic stem cell maintenance" for consideration by *eLife*. Your article has been reviewed by 3 peer reviewers, one of whom is a member of our Board of Reviewing Editors, and the evaluation has been overseen by Richard White as the Senior Editor. The reviewers have opted to remain anonymous.

Essential revisions:

1) The authors need to formally exclude the possibility that expression of the small nucleolar genes Snord100 and Snora33, located in Rps12 introns 4 and 5, respectively, is also perturbed by the deletion of Rps12 exon 2. Any change in the expression of these snoRNAs would confound the interpretation that all the described phenotypes map back to Rsp12 specifically. Along these lines, the authors should also provide data to assess the impact of the Rps12 mutation on Rps12 protein expression.

2) Northern blots and/or other molecular assays should be used to characterize the extent to which haploinsufficiency of RpS12 impacts overall ribosome biogenesis. In addition, polysome profiles in the Rps12 mutant animals should be analyzed to assess 40S/60S subunits, 80S monosomes and polysomes.

3) The authors need to devise experiments to distinguish between direct and indirect consequences of Rps12 loss. For example, the authors should examine the extent to which acute loss of Rps12 regulates HSPCs, ribosome biogenesis and assembly, and ribosome composition. Cell proliferation, survival, and protein synthesis rate assays require fewer cells than polysome profiling, and these experiments would begin to provide insights into the direct cellular consequences of Rps12 haploinsufficiency.

These three points cover many of the shared concerns of the reviewers. The authors should do their best to address the individual critiques of each of the reviewers (see below).

*Reviewer #1 (Recommendations for the authors):*

1) The expression of both snoRNAs should be evaluated in the homozygous and heterozygous mutants. The expression levels of Rps12 protein should also be determined in the heterozygous mutants.

2) Rescue of Rps12 mutant phenotypes by a Rps12 transgene should be conducted, given the probability that the expression of the snoRNAs will be disrupted in the mutant.

3) Northern blots and/or other molecular assays should be used to characterize the extent to which haploinsufficiency of RpS12 impacts overall ribosome biogenesis.

4) Have the authors considered the possibility that loss of Rps12 triggers a feedback loop that promotes ribosome biogenesis? The authors could evaluate nascent rRNA transcription levels as one measure of how the loss of this RP gene impacts ribosome homeostasis.

*Reviewer #2 (Recommendations for the authors):*

Key points

1. Have the authors formally excluded the possibility that expression of the small nucleolar genes Snord100 and Snora33, located in Rps12 introns 4 and 5, respectively, is also perturbed by the deletion of Rps12 exon 2 i.e. can they be sure that all the phenotypes are indeed Rps12-dependent?

2. Please provide data to assess the impact of the Rps12 mutation on Rps12 protein expression.

3. It is important to analyze the potential functional consequences of Rps12 deletion on ribosomal subunit assembly. Polysome profiles in the Rps12 mutant animals should be analyzed to assess 40S/60S subunits, 80S monosomes and polysomes. Is there any effect of Rps12 deletion on rRNA processing?

4. To what extent are the observed Rps12 mutant phenotypes indirect consequences of TP53 activation? What is the impact of the Rps12 mutation on TP53 stabilization in the different tissues affected by the mutation?

*Reviewer #3 (Recommendations for the authors):*

A few key experiments may help elucidate the mechanisms by which Rps12 regulates HSPCs.

1. Since this is a floxed allele, it is possible to induce acute loss of Rps12 (or Rps f/+) in vivo or ex vivo in HSPCs, to study the impact of the acute loss of Rps12. Does Rps12 haploinsufficiency reduce protein synthesis or really increase protein synthesis as shown in this manuscript?

2. Following up on point #1 experiment, polysome profiling will further dissect the role of Rps12 in ribosome biogenesis and assembly.

3. Has this (polysome profiling and signaling transduction) been done in hematopoietic cell line with Rps12 knockdown or CRISPR-mediated deletion of one allele of Rps12? If not, the authors should investigate the role of Rps12 in mammalian hematopoietic cell lines.

4. It is important to devise experiments to distinguish the direct and indirect consequences of Rps12 loss. The ERK/mTOR signaling activation and increases in protein synthesis were consequences, or negative feedback of Rps12 loss?

5. It is important to devise experiments to distinguish the ribosomal and extraribosomal function of Rps12 in mammalian cells.

6. The phenotypes presented in this manuscript showed a strong HSPCs and pan-hematopoietic effects. Erythropoiesis was actually less affected. The narrative and discussion should reflect that, and de-emphasize its role in or similarity to DBA. In particular, RPS12 is not found mutated in DBA? Thus, this mouse model is not really relevant to DBA.

7. It is interesting that Rps12 haploinsufficiency does not affect fetal liver hematopoiesis or erythropoiesis, but strongly impacts this process in adult. The explanation of this phenotype may not have anything to do with HSPC cycling rate difference between fetal liver versus bone marrow HSPCs, as speculated in the Discussion. It is important to note that a number of other mouse models such as MplKO mice, also have little phenotypes in fetal liver HSCs but strong phenotypes in adult bone marrow HSCs. Fetal liver and adult bone marrow HSPCs have different transcriptional programs, no coding RNAs, and epigenetic landscapes.

---

## [Author Response]

Essential revisions:1) The authors need to formally exclude the possibility that expression of the small nucleolar genes Snord100 and Snora33, located in Rps12 introns 4 and 5, respectively, is also perturbed by the deletion of Rps12 exon 2. Any change in the expression of these snoRNAs would confound the interpretation that all the described phenotypes map back to Rsp12 specifically. Along these lines, the authors should also provide data to assess the impact of the Rps12 mutation on Rps12 protein expression.

We found that the RpS12 excision allele affected the expression of RpS12 mRNA and protein, but that levels of Snord100 and Snora33 were not changed. These results, which are consistent with the idea that the phenotype is caused by RpS12 haploinsufficiency, are described in lines 236-244 of the manuscript, and the data also shown in the itemized response to individual critiques further below.

2) Northern blots and/or other molecular assays should be used to characterize the extent to which haploinsufficiency of RpS12 impacts overall ribosome biogenesis. In addition, polysome profiles in the Rps12 mutant animals should be analyzed to assess 40S/60S subunits, 80S monosomes and polysomes.

We used qRT-PCR to find that overall ribosome subunit numbers were not detectably altered by the mutation. We did not investigate ribosome biogenesis or polysome profiles in more detail, because it is reported in yeast that RpS12 deficiency does not affect ribosome assembly and that otherwise mature SSU lacking RpS12 appear in the cytoplasm, and because the revised conclusion that RpS12 haploinsufficiency results primarily in a reduction in global translation, not an increase (see point 3 below), no longer suggests a unique mode of translation. In any case, it would be technically difficult to perform polysome profiling from the reduced numbers of stem and progenitor cells available in the mutant. These findings and their discussion are reported in lines 236-244 of the revised manuscript, and also shown in the itemized response to individual critiques further below.

3) The authors need to devise experiments to distinguish between direct and indirect consequences of Rps12 loss. For example, the authors should examine the extent to which acute loss of Rps12 regulates HSPCs, ribosome biogenesis and assembly, and ribosome composition. Cell proliferation, survival, and protein synthesis rate assays require fewer cells than polysome profiling, and these experiments would begin to provide insights into the direct cellular consequences of Rps12 haploinsufficiency.

As suggested, the revised manuscript includes acute loss of RpS12 from adult hematopoietic cells, with the conclusion that the primary effect of RpS12 haploinsufficiency is reduced translation, followed by cytopenia, and that stem cell reactivation with elevated translation is a more complex response to chronic and early RpS12 depletion. These experiments are described in lines 371-393 of the revised manuscript, the results also shown in the itemized response to individual critiques further below, and represent the major effort undertaken to revise the manuscript.

Reviewer #1 (Recommendations for the authors):1) The expression of both snoRNAs should be evaluated in the homozygous and heterozygous mutants. The expression levels of Rps12 protein should also be determined in the heterozygous mutants.

Supplementary Figure 3A-C of the revised manuscript shows that snoRNA levels are unaffected in heterozygous mutants, whereas RpS12 mRNA and protein levels are reduced.

Gene expression levels cannot be examined in homozygous mutants, since none are recovered.

2) Rescue of Rps12 mutant phenotypes by a Rps12 transgene should be conducted, given the probability that the expression of the snoRNAs will be disrupted in the mutant.

Please see point 1 above. In fact, snoRNA levels are unaffected in heterozygous mutants, whereas RpS12 mRNA and protein levels are reduced.

3) Northern blots and/or other molecular assays should be used to characterize the extent to which haploinsufficiency of RpS12 impacts overall ribosome biogenesis.

Supplementary Figure 3D-D’ of the revised manuscript shows Bioanalyzer data indicating that steady state levels of small subunits are not changed in the heterozygous mutant, nor is the SSU/LSU ratio. This is consistent with prior studies in yeast, which showed that RpS12 deficiency does not affect SSU biogenesis. These results are described in lines 242-244 of the manuscript, and discussed in lines 433-441 of the manuscript.

4) Have the authors considered the possibility that loss of Rps12 triggers a feedback loop that promotes ribosome biogenesis? The authors could evaluate nascent rRNA transcription levels as one measure of how the loss of this RP gene impacts ribosome homeostasis.

We did not investigate this possibility, as we now conclude that RpS12 haploinsufficiency initially reduces overall translation, rather than increasing it.

Reviewer #2 (Recommendations for the authors):Key points1. Have the authors formally excluded the possibility that expression of the small nucleolar genes Snord100 and Snora33, located in Rps12 introns 4 and 5, respectively, is also perturbed by the deletion of Rps12 exon 2 i.e. can they be sure that all the phenotypes are indeed Rps12-dependent?

Yes, as reported in the response to reviewer 1, point 1, above.

2. Please provide data to assess the impact of the Rps12 mutation on Rps12 protein expression.

We have examined Rps12 protein expression in Rps12 KO/+ ckit+ bone marrow cells by Western analysis, and we observed a trend towards decrease in Rps12 protein levels in Rps12 heterozygous KO cells (Supplementary figure 3C) and addressed it in the text in lines 240-242.

3. It is important to analyze the potential functional consequences of Rps12 deletion on ribosomal subunit assembly. Polysome profiles in the Rps12 mutant animals should be analyzed to assess 40S/60S subunits, 80S monosomes and polysomes. Is there any effect of Rps12 deletion on rRNA processing?

Please see data provided in response to reviewer 1, point #3. We did not follow up with polysome profiles. This is of less interest now given the conclusion of the revised manuscript that RpS12 haploinsufficiency reduces overall translation. Polysome profiling of mutant stem and progenitor cells would be very challenging, as their numbers are much reduced.

4. To what extent are the observed Rps12 mutant phenotypes indirect consequences of TP53 activation? What is the impact of the Rps12 mutation on TP53 stabilization in the different tissues affected by the mutation?

The TP53 antibodies we tried gave poor results and in the interests of time we did not continue this further. We request that this be considered beyond the scope of the current study.

Reviewer #3 (Recommendations for the authors):A few key experiments may help elucidate the mechanisms by which Rps12 regulates HSPCs.1. Since this is a floxed allele, it is possible to induce acute loss of Rps12 (or Rps f/+) in vivo or ex vivo in HSPCs, to study the impact of the acute loss of Rps12. Does Rps12 haploinsufficiency reduce protein synthesis or really increase protein synthesis as shown in this manuscript?

We thank the reviewer for this excellent suggestion, which we believe has really enhanced the manuscript (please see lines 371-393 and 436-445). To determine the direct acute effects of Rps12 deletion specifically in hematopoietic cells, we generated a new RpS12flox/+; SCl^-^CreER mouse model which allowed us to acutely delete the RpS12 in the bone marrow. This new data is included in the new Figure 8 and Suppl Figure 4., and in the text in the Results section on p16 lines 367 and in the Discussion on page 20, line 507-522. We observed that it takes some time to develop the loss of HSC maintenance and pancytopenia phenotype after Rsp12 hetorozygous excision. Interestingly, at 12-weeks post RpS12 deletion in the bone marrow, there were no changes in peripheral blood counts, total bone marrow cellularity, or in the bone marrow stem and progenitor cell populations (Figure 8C). However, at that timepoint we already observed a significant decrease in OPP incorporation in hematopoietic stem and progenitor cells, consistent with decreased translation (Figure 8D). However, there were no significant differences in the cell cycle profile of these HSPCs at that time point (Supp Figure 2D). At 14-weeks post *RpS12* deletion, we started to observe mild pancytopenia (Figure 8E) and a significant decrease in bone marrow cellularity, along with a decrease in HSC numbers and a trend toward decrease in progenitor populations (Figure 8F). These data suggest that the immediate effect of acute loss of RpS12 in adult hematopoietic cells is actually decreased translation, leading to a decrease in HSCs, BM cellularity and pancytopenia. This also suggests that the increased cell cycling, AKT/ERK/MTOR signaling, and translation that we observed in Rps12 KO/+ mice with embryonic deletion were likely compensatory to this loss in HSC maintenance.

2. Following up on point #1 experiment, polysome profiling will further dissect the role of Rps12 in ribosome biogenesis and assembly.3. Has this (polysome profiling and signaling transduction) been done in hematopoietic cell line with Rps12 knockdown or CRISPR-mediated deletion of one allele of Rps12? If not, the authors should investigate the role of Rps12 in mammalian hematopoietic cell lines.

We did not perform polysome profiles. Please see the response to Reviewer 1, point #3.

4. It is important to devise experiments to distinguish the direct and indirect consequences of Rps12 loss. The ERK/mTOR signaling activation and increases in protein synthesis were consequences, or negative feedback of Rps12 loss?

Based on our new model we think that the direct effect of Rps 12 loss in hematopoietic cells is decreased translation, leading to impaired HSC maintenance (see the response to point 1 and results in Figures 8 and Supplementary Figure 4). Therefore, we think that the increased translation that was observed in Rps12 KO/+ mice with embryonic deletion was compensatory due to loss of HSCs. We discuss our current understanding in lines 441-445 of the revised manuscript.

5. It is important to devise experiments to distinguish the ribosomal and extraribosomal function of Rps12 in mammalian cells.

This is an important issue but not easily addressed. This has not been addressed in mouse models of other Rp mutants either, so there is no experimental road map to follow. We suggest it is beyond the scope of the current paper.

6. The phenotypes presented in this manuscript showed a strong HSPCs and pan-hematopoietic effects. Erythropoiesis was actually less affected. The narrative and discussion should reflect that, and de-emphasize its role in or similarity to DBA. In particular, RPS12 is not found mutated in DBA? Thus, this mouse model is not really relevant to DBA.

RPS12 might be mutated or deleted in DBA patients for which the causative mutation has not yet been found. RPS12 has not been included in any standard NGS panels for bone marrow failure syndromes. We think our report of hematopoietic phenotypes in Rsp12 KO mice should prompt the inclusion of RPS12 in future bone marrow failure NGS panels. We discuss this in lines 446-460.

7. It is interesting that Rps12 haploinsufficiency does not affect fetal liver hematopoiesis or erythropoiesis, but strongly impacts this process in adult. The explanation of this phenotype may not have anything to do with HSPC cycling rate difference between fetal liver versus bone marrow HSPCs, as speculated in the Discussion. It is important to note that a number of other mouse models such as MplKO mice, also have little phenotypes in fetal liver HSCs but strong phenotypes in adult bone marrow HSCs. Fetal liver and adult bone marrow HSPCs have different transcriptional programs, no coding RNAs, and epigenetic landscapes.

Thanks to the reviewer for raising this point, we have removed the discussion about differences between the cycling of adult and fetal liver HSCs from the discussion.